RESEARCH

# An integrated multi-omics approach to identify regulatory mechanisms in cancer metastatic processes

Saba Ghaffari[1] , Casey Hanson[2], Remington E. Schmidt[3], Kelly J. Bouchonville[3], Steven M. Offer[3*] and Saurabh Sinha[4*]

\* Correspondence: Offer.Steven1@
mayo.edu; sinhas@illinois.edu
[3]Department of Molecular
Pharmacology and Experimental
Therapeutics, Mayo Clinic, Gonda
19-476, 200 First St SW, Rochester,
MN 55905, USA
[4]Department of Computer Science,
Carl R. Woese Institute of Genomic
Biology, and Cancer Center of
Illinois, University of Illinois at
Urbana-Champaign, 2122, Siebel
Center, 201 N. Goodwin Ave.,
Urbana, IL 61801, USA
Full list of author information is
available at the end of the article

## Abstract

**Background:** Metastatic progress is the primary cause of death in most cancers, yet the regulatory dynamics driving the cellular changes necessary for metastasis remain poorly understood. Multi-omics approaches hold great promise for addressing this challenge; however, current analysis tools have limited capabilities to systematically integrate transcriptomic, epigenomic, and cistromic information to accurately define the regulatory networks critical for metastasis.

**Results:** To address this limitation, we use a purposefully generated cellular model of colon cancer invasiveness to generate multi-omics data, including expression, accessibility, and selected histone modification profiles, for increasing levels of invasiveness. We then adopt a rigorous probabilistic framework for joint inference from the resulting heterogeneous data, along with transcription factor binding profiles. Our approach uses probabilistic graphical models to leverage the functional information provided by specific epigenomic changes, models the influence of multiple transcription factors simultaneously, and automatically learns the activating or repressive roles of *cis*-regulatory events. Global analysis of these relationships reveals key transcription factors driving invasiveness, as well as their likely target genes. Disrupting the expression of one of the highly ranked transcription factors JunD, an AP-1 complex protein, confirms functional relevance to colon cancer cell migration and invasion. Transcriptomic profiling confirms key regulatory targets of JunD, and a gene signature derived from the model demonstrates strong prognostic potential in TCGA colorectal cancer data.

**Conclusions:** Our work sheds new light into the complex molecular processes driving colon cancer metastasis and presents a statistically sound integrative approach to analyze multi-omics profiles of a dynamic biological process.

**Keywords:** Colon cancer, Metastasis, Transcriptional regulation, Multi-omics, Probabilistic model

## Background

Globally, colorectal cancer (CRC) has the third highest incidence and second highest rate of cancer-related deaths [1]. Progression from primary cancer to metastatic disease is the most common cause of mortality in solid malignancies such as CRC, and approximately half of CRC cases will either present as metastatic disease or develop metastases regardless of cancer treatment [2]. While specific driver mutations are well-defined in CRC oncogenesis, the mechanisms that facilitate metastatic progression are poorly understood. Changes in gene expression have been shown to accompany CRC progression and to predict metastasis [3–5]. Epigenetic changes have also been associated with CRC pathogenesis [6–8]; however, current analysis tools offer limited ability to integrate transcriptome and epigenome data to precisely define the regulatory frameworks of metastasis.

Here, we approached this problem using a purposefully generated cellular model of CRC invasiveness (a hallmark of metastasis). Using multi-omics profiling of cells at increasing levels of invasiveness and a novel integrative analysis framework yielded several novel insights about transcriptional regulatory mechanisms underlying the transcriptomic dynamics of CRC progression.

We profiled gene expression, as well as genome-wide profiles of DNA accessibility and four select histone modifications known to be associated with *cis*-regulatory information, in four different stages of progression. These data allowed us to identify large numbers of genes that change the expression in either direction as the cell populations acquire more invasive characteristics, and the genome-wide epigenomic profiles yielded many potential *cis*-regulatory regions associated with those changes. However, this in itself does not reveal details of the transcriptional regulatory network (TRN), i.e., the specific transcription factors (TFs) and TF-gene relationships that drive the transcriptomic changes and are reflected in the *cis*-regulatory regions. We therefore combined the above data with genome-wide colon cancer cell line TF-DNA binding profiles from the ENCODE Project. By combining TF-binding site (TFBS) information with epigenomics-based markers of *cis*-regulatory segments and differential expression of nearby genes, we were able to identify the TFs most likely to regulate CRC progression, as well as their putative target genes. The strategy of finding statistical enrichments of a TF's binding sites in the regulatory regions associated with differentially expressed genes is a time-tested one [9–11]; here, we hoped to significantly improve its efficacy by additionally using epigenomic data from the cellular contexts being contrasted.

A key aspect of our strategy was the use of changes in histone modifications between stages. Specific histone modifications have been associated with activating or repressive influences [12–14] on the gene expression, so one expects improved regulatory analysis by focusing on TFBS that are flagged by such marks. Moreover, some previous studies have argued that *changes* in epigenomic state provide valuable information about regulatory mechanisms underlying cellular state transitions, perhaps more so than merely the presence or absence of epigenomic marks. For instance, Bozek et al. reported accessibility of *cis*-regulatory elements to vary along the antero-posterior axis in *Drosophila* blastoderm, in a manner correlated with their regulatory activity [15]. Thus, we focused on TFBS that coincide with dynamic histone marks rather than simply the presence of marks.

Another challenge we were faced with pertains to the use of regulatory *direction*, i.e., activating or repressive influence, associated with specific histone modifications. For

instance, it seems natural to focus on TFBS flagged by an activating mark such as H3K27ac when located near an upregulated gene. However, one might argue that such a TFBS if located near a downregulated gene presents inconsistent information about the TF's regulatory influence, especially if the TF is known to be an activator. This point is even more germane if our analysis is based on dynamic histone marks: for example, when seeking evidence of an activator TF regulating a gene that is upregulated in later (more invasive) stages, one should consider TFBS flagged by an increase in an activating histone mark or decrease in a repressive histone mark, with either epigenomic change pointing to a more activating chromatin context in the later stages. As this illustrative scenario suggests, our analysis needs to account for regulatory directions associated with TFs, epigenomic marks, and differential expression, in order to narrow down the large numbers of putative *cis*-regulatory elements to those most likely to be functional. Furthermore, the regulatory directions of TFs are seldom known, and even those of specific histone marks are not always well understood; hence, we sought these biological characteristics to be automatically learnt from data.

It is well known that genome-wide binding sites of different TFs often exhibit high degrees of co-localization [16], e.g., due to frequent TF binding at accessible regions of DNA, and large numbers of TFBS do not have the obvious regulatory function expected of them [17]. One strategy to mitigate the resulting problems in the statistical approach noted above is to analyze the associations of differentially expressed genes with many or all TFs concurrently rather than test enrichments for each TF separately. In a related but different context, previous studies have utilized multi-TF modeling of genes to discover TRNs from the expression data [18–20]. Inspired by these studies, we developed here an analogous multi-TF model of gene expression to discover TF-gene regulatory relationships based on TF-DNA binding and epigenomic evidences.

Our analytical framework uses a rigorous probabilistic model to integrate gene expression and epigenomic data from different cellular states (maternal and invasive cell lines) with TF-DNA binding data from a related cell line, to identify TFs that regulate the observed transcriptomic dynamics. The model automatically learns dominant regulatory directions associated with each TF and histone mark for which data are available and also predicts the likely target genes of each TF. Using rigorous statistical evaluations, we showed that the use of dynamic histone marks has significant advantages over simpler strategies that do not fully exploit this rich source of *cis*-regulatory information. The model predicted several important regulatory pathways that are commonly associated with oncogenic and metastatic phenotypes, including the AP-1 complex members JunD and Fosl. We experimentally tested the role of JunD by shRNA-mediated knockdown and found the resulting cell line to exhibit significantly reduced migration and invasion characteristics. RNA-seq profiling of the JunD knockdown condition revealed a large set of potential targets of this TF. We found this set to be significantly enriched for model-based predictions of JunD targets, thereby confirming our ability to infer TF-gene relationships. Finally, we constructed a gene signature of CRC invasiveness based on predicted targets of the most significant TFs and showed that this signature has stronger prognostic value for predicting the overall survival in CRC than gene expression alone. In summary, we present here a multi-omics, statistically rigorous strategy to investigate the *cis*-regulatory mechanisms underlying a complex biological process and use it to glean new insights into colorectal cancer progression.

## Results

### Multi-omics profiling of a CRC cell line

Acquiring the ability to migrate and invade through host tissues is a hallmark of metastatic cancer cells. To identify differentially regulated pathways in this process, matched SW480 cell models with varying degrees of invasiveness were derived by repeated selection of cells capable of chemotaxis through a microporous membrane coated with Matrigel extracellular matrix (Fig. 1a). In the subsequent sections of the text, the number of times cells were selected using Matrigel-coated membranes is denoted as the "M" number for the cell line, where M0 is the parental SW480 culture that has not undergone selection, M1 is a culture that has been selected one time, and so on. Two completely independent biological replicate series of cultures were derived using the same methodology by two independent researchers.

Cells that had undergone repeated rounds of selection displayed increased invasiveness (Additional file 1: Figure S1). To identify gene expression changes in invasive cells, mRNA sequencing (RNA-seq) was performed on RNA harvested from cultures M0, M2, M4, and M6. Principal component analysis (PCA) indicated that profiles for M4 and M6 are distinct from those for M0 and M2 (Additional file 1: Figure S2), but profiles for M4 and M6 were more similar within replicates than by stage (Additional file 1: Figure S2). Because of the clear expression and phenotypic separation between M0 and M6, further analyses focused on those cell lines, beginning with characterization of differentially expressed (DE) genes (Additional file 1: Figure S3). Gene set characterization of the DE genes (adjusted *p*-value ≤ 0.05) using the KnowEnG system [21] revealed several physiologically relevant properties of these genes (see Additional file 2). The down-regulated genes (adjusted *p*-value ≤ 0.05) showed a strong enrichment for cancer-related gene signatures from mSigDB [22], most notably E-cadherin (*CDH1*) targets (hypergeometric test *p*-value 3.7E−41), the loss of which is a generalized hallmark of metastatic cells that have undergone epithelial-mesenchymal transition (EMT). Additional dysregulated pathways include those associated with invasion/migration in varied cancer types, including metastasis in melanoma (*p*-value 6.2E−15) and migration in bladder cancer cell lines (*p*-value 9.7E−14). Upregulated mSigDB modules include pathways typically associated with breast cancer invasiveness, including SMARCE1 targets (*p*-value 2.6E−16), ESR1 targets (*p*-value 5.3E−16), and a comparison of luminal and mesenchymal breast cancers (*p*-value 2.1E−12). While these pathways are often associated with breast cancer, there is also precedence for general cancer relevance. For example, SMARCE1 is a core subunit of the SWI/SNF chromatin remodeling complex that has been linked to invasiveness in a hormone-independent manner in additional cancers [23].

To determine if the changes observed between M0 and M6 could be attributed to the selection of a specific genetic sub-population of cells, we performed variant calling on RNAseq data from M0 and M6 cells. A subset of variant loci with high depth across M0 and M6 lines was selected to assess shifts in population allele frequencies between M0 and M6 as a measure of enrichment (Additional file 1: Figure S4). Notably, allele frequencies are largely similar between stages, with a relatively small number of exceptions (points along the axes in Additional file 1: Figure S4), indicating that genetic cell identity remains consistent from M0 to M6. Additionally, no obvious driver mutations associated with colorectal cancer progression were noted (Additional file 3).

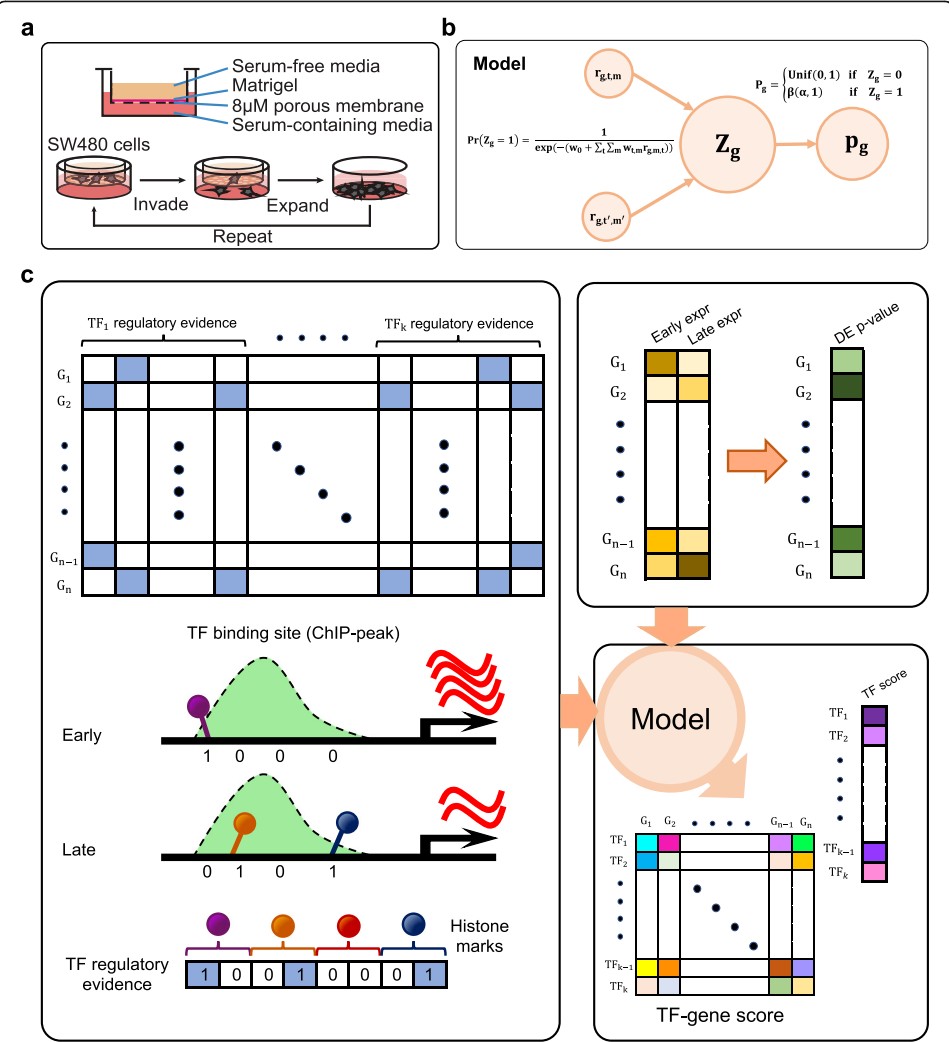

**Fig. 1** Schematics of study design and analysis framework. **a** An invasive sub-culture of SW480 cells was established by repeated selection of cells that could invade through porous membrane coated with synthetic extra-cellular matrix toward a chemoattractant (serum). **b** The pGENMi probabilistic model was adapted to aggregate *cis*-regulatory evidence associated with each differentially expressed (DE) gene. $P_g$ represents the differential expression *p*-value of gene *g*, $Z_g$ is a binary hidden variable that represents if *g* mediates the regulatory influence of one or more known TFs on CRC invasiveness, and $r_{g,t,m}$ represents a (binary) *cis*-regulatory evidence in the form of a binding site for TF *t*, flagged by dynamic histone mark *m*, in the regulatory region of gene *g*. The weighted sum of *cis*-regulatory evidence (with learnable weights $w_{t,m}$) determines $\Pr(Z_g = 1)$. $P_g$ follows a beta distribution if $Z_g = 1$ and is uniform if $Z_g = 0$. **c** Overview of the analysis. Left panel depicts the matrix of *cis*-regulatory evidence for multiple TFs and all genes. A TFBS overlapping with a change of histone mark between stages is encoded with two bits, one for either direction of change. Each TF is thus represented by eight bits, representing four histone marks. The evidence matrix and the DE *p*-values of genes (between the early and late stages) are inputs to the model. The output of the model contains a score assigned to each TF representing its contribution to the model and a score associated with each (TF, gene) pair representing the extent to which the gene mediates the effect of the TF on CRC invasiveness

To understand the regulatory mechanisms underlying the differential expression of genes between stages, we performed genome-wide ChIP-seq profiling of four different histone modifications—H3K27ac, H3K4me1, H3K4me3, and H3K27me3—as well as genome-wide ATAC-seq to profile DNA accessibility, in the early as well as late stages. We first examined the global changes in histone marks and chromatin accessibility by summarizing how counts of histone mark ChIP peaks and DNA accessibility peaks

change across stages. Specifically, we counted the peaks within 10 kbp upstream of the genes for each stage of progression, limiting ourselves to genes that are differentially expressed (*p*-value < 0.05) between early (M0) and late (M6) stages. This was done for up- and downregulated genes separately. The results (Additional file 1: Figure S5) show clear trends of genome-wide epigenomic changes. For instance, H3K27ac peaks near downregulated genes are fewer in later stages, and those near upregulated genes are more numerous in later stages, as might be expected of an activating histone mark. The reverse pattern exists for H3K27me3 peaks, consistent with a repressive role for this mark. Similar trends were observed in the changes of signal strength between stages (Additional file 1: Figure S6).

### Integrative analysis of expression, TF binding, and epigenomic data: outline of model

A common and simple approach to regulatory analysis is to ask if a TF's binding sites enrich near DE genes [10]. We obtained ChIP-seq profiles of genome-wide DNA binding for 20 TFs in the colon cancer cell line HCT116 (Additional file 4: Table S1). ChIP peaks from these data provide us with putative TFBS associated with each gene, allowing enrichment tests to be performed. However, TFBS from ChIP-seq experiments are known to be promiscuous and a poor predictor of functional TF-gene relationship [17, 24]. As a result, the baseline strategy of testing TFBS enrichments in the gene regulatory regions is typically confounded by a high rate of false-positive sites. This issue is exacerbated by searching over longer intergenic regions with the intent to identify more sensitive TFBS-gene associations. Our epigenomic profiles can mitigate this problem by increasing the functional specificity of the TFBS information. For instance, we may only consider those TFBS that are located within active enhancers as indicated by specific histone marks, thus increasing the specificity of *cis*-regulatory evidence of a TF regulating a gene. We sought to further increase the specificity of *cis*-regulatory evidence by considering the *changes* in the epigenomic state [25] by using changes in histone marks between stages as a filter for TFBS and subsequent testing for enrichment of a TF's binding sites near DE genes.

  As noted above, data were generated for four different histone marks, one or more of which may contribute to the inter-stage changes in the gene expression and furnish, perhaps with different specificities, *cis*-regulatory evidence for TF-gene relationships. Therefore, our approach was to utilize all four studied histone marks simultaneously in our integrative model. We interpreted the changes in multiple histone marks at a putative TFBS as a stronger evidence of the TF's influence than that provided by a single type of epigenomic evidence. In considering changes in histone marks, we assumed that the direction of change is informative, e.g., a histone mark that appears near a gene only in the late stage should have either an activating or repressing role, and this role should remain consistent across genes. However, we did not assume knowledge of such roles a priori; the data furnished this information. We also allowed different types of dynamic histone marks to have different evidentiary values. For example, a TFBS that overlaps a change in H3K27ac might be more reliable evidence of the TF's regulatory influence compared to an overlap with a change in H3K4me1. Also, as noted in the introduction, we recognize that the differential expression of genes between stages is likely under the regulatory control of multiple TFs. Hence, we analyzed associations between DE genes and all TFs in a multi-TF model rather than one TF at a time.

We built upon our previously published pGENMi model [26] to analyze the multi-omics data (Fig. 1c; detailed in the "Methods" section). To set up the model, each gene was associated with a differential expression $p$-value and a set of binary evidences for TF regulatory influence. Each binary evidence corresponds to a pair (T, M) of TF (T) and change in a specific histone mark (M; e.g., an H3K27ac peak exclusive to the late stage). The binary evidence is true if a ChIP peak for a TF T overlaps the dynamic histone mark M, within a pre-determined distance $d$ from the gene's start site, and false otherwise (Fig. 1c). Since there are four histone marks in our data, and each mark may change in one of two directions, there are eight binary evidences for each TF, for a total of 160 evidences representing 20 TFs. The model uses a hidden binary variable $Z_g$ for each gene $g$, representing whether or not the gene's differential expression is associated with one or more of the TFs (Fig. 1b). The probability of $Z_g = 1$ is a logistic function of the weighted sum of all binary evidences available for it, i.e., one or more TF-binding sites near the gene, each supported by a dynamic histone mark, makes the gene more likely to be a target of those TFs (Fig. 1c). Moreover, the observed DE $p$-value of the gene is modeled by two different probability distributions depending on whether $Z_g = 1$ or $Z_g = 0$, with the former case ($Z_g = 1$) creating a bias toward small $p$-values (Fig. 1b). As a result, the likelihood of the data is higher if there are many genes for which the DE $p$-value is small (significant), and such genes have one or more regulatory evidences associated with them. The weights of binary evidences determining $\Pr(Z_g = 1)$ are free parameters ($w_{T,M}$) (Fig. 1b) learned from the data by maximum likelihood, and regularization is used to avoid overfitting. To achieve consistency between direction of differential expression and the regulatory direction of TFs and histone mark changes, we performed the entire analysis (model training) twice, with DE $p$-values representing the significance of upregulation and downregulation respectively. These two analyses are henceforth referred to as up- and down-analysis.

### Identification of transcription factors underlying CRC invasiveness

We learned the optimal values of the model's hyperparameters—distance threshold ($d$ = 10 kbp, 50 kbp, 200 kbp, or 1 Mbp) and regularization coefficient—by cross-validation on the entire dataset (Fig. 2a, Additional file 1: Figure S7). Here, all genes were randomly partitioned into training (80%) and test sets (20%), and model accuracy was evaluated by log likelihood ratio (LLR) on the test genes. The cross-validation was performed for up- and down-analyses separately, and the overall test accuracy for each setting of the two hyperparameters was measured by the sum of test LLRs in these two analyses, averaged over 100 repeats of random cross-validation. This identified 50 kbp as the optimal distance threshold for *cis*-regulatory evidence, though similar accuracy values were noted for the shorter range of 10 kbp and the greater range of 200 kbp. The model was found to perform significantly worse when using a regulatory region of 1 Mbp upstream or downstream of the gene, which suggests that considering TFBS at great distances (e.g., over 200 kbp from a gene), even with the support of epigenomic information, potentially includes more noise than signal in our analysis.

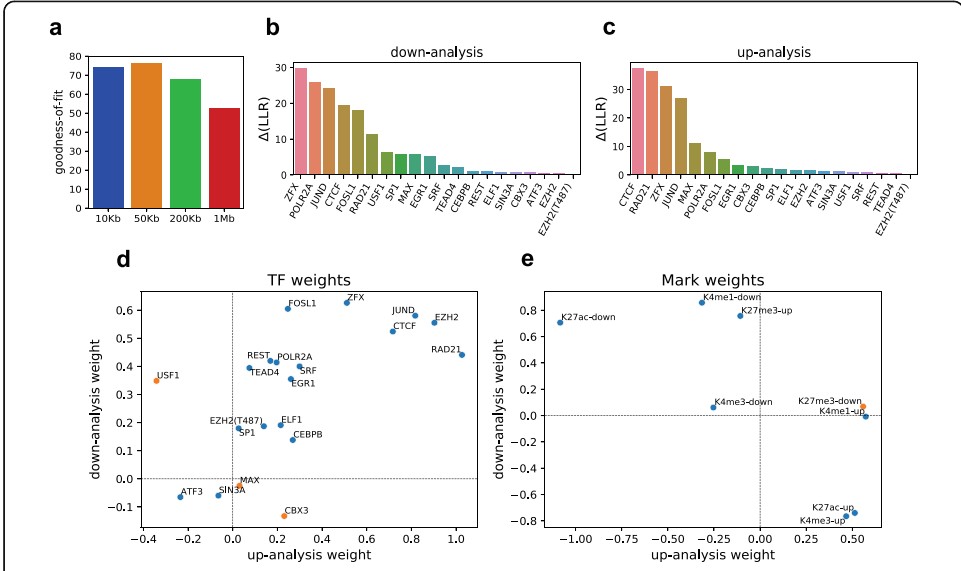

**Fig. 2** Regulatory influences learnt by model. **a** Comparison of goodness-of-fit for each distance threshold (maximum distance upstream or downstream of gene) used for associating TF ChIP-peaks with genes. The goodness-of-fit is measured by the sum of test LLRs (derived from cross-validation on the entire dataset) from up- and down-analyses, averaged over 100 repeats of the procedure. The distance and regularization coefficient used by the best-fit model were then used to re-train the model on the entire dataset. **b**, **c** Model-based ranking of TFs, in down- and up-analysis, respectively. Each TF's contribution was measured by zeroing its regulatory evidence and calculating the change in model LLR ($\Delta(LLR)$). **d** TF weights learned by the fw-pGENMi for down-analysis and up-analysis. All of the TFs except three, USF1, MAX, and CBX3, were assigned a consistent role in both analyses. A positive weight suggests an activating role for a TF while a negative weight represents a repressive role. **e** Weights for histone mark changes learned by fw-pGENMi for down- and up-analysis. A positive weight for a "mark-up" (respectively, "mark-down") change for up-analysis (respectively, down-analysis) suggests an activating histone mark. This is the case for H3K27ac, H3K4me1, and H3K4me3. The mark H3K27me3 has the opposite pattern, consistent with a repressive role

Using the optimal hyperparameter values obtained above, we re-trained the model on the entire dataset (all genes) and ranked TFs by their contribution to the model, separately for the down- and up-analyses (Fig. 2b, c). We did this by zeroing out all binary evidence related to a TF (one TF at a time), recalculating the model LLR, and using the difference of LLRs, called $\Delta$LLR, before and after zeroing the TF's regulatory evidence, as the contribution of that TF. Both analyses identified ZFX, JUND, and CTCF in the top 5 ranked TFs. Additionally, RAD21 (ranks 6 and 2) and FOSL1 (ranks 5, 7) also consistently ranked highly (we present a more rigorous assessment of the statistical significance of these TFs below). A direct look at the binding sites of one of these TFs, viz., JunD (Additional file 1: Figure S8), shows substantial epigenomic changes in both directions. However, it does not immediately offer a mechanistic explanation of such changes or a quantitative assessment of their impact on gene expression and illustrates the need for a more nuanced analysis cognizant of expression changes, as is provided by our model.

ZFX is a transcriptional activator of that has been linked to oncogenic processes in numerous cancer types [27] and has been correlated with aggressive tumor phenotypes and poor survival in colorectal cancer [28]. JUND and FOSL1 are both potential components of the dimeric AP-1 transcription factor complex. AP-1 transcription factor complexes are generally considered oncogenic; however, the specific contributions to

cancer development and progression can be dependent upon the dimeric composition of AP-1 transcription factors, cell type, tumor stage, and genetic background [29]. CTCF is a multi-functional protein that can act as a transcriptional activator, transcriptional repressor, or an insulator element [30]. CTCF-regulated genes are strongly enriched in cancer-related pathways, including cell differentiation, proliferation, viability, migration, and adhesion [31]. While there is considerable literature evidence that these highly ranked TFs are involved in cancer-related processes, it is also appreciated that the TFs evaluated by the ENCODE Project were likely enriched for those relevant to human disease, including cancer.

### Model reveals regulatory roles of transcription factors and histone marks

To probe the inner workings of the model, we next trained a modified version of the pGENMi model, henceforth called "factorized weights pGENMi" (fw-pGENMi). In the model trained above, every combination of TF and dynamic histone mark was considered a separate evidence type and had an associated parameter $w_{T,M}$. Suppose we are analyzing the upregulation of genes (up-analysis), and this parameter is learnt to be positive. This means that histone mark change "M" (e.g., appearance of H3K27ac mark in the late stage) at a binding site for TF "T" (e.g., JunD) is an evidence of upregulation of the gene. Similarly, if the parameter is fit to a negative value, it means that the histone mark change "M" at a binding site of TF "T" is suggestive of the downregulation of the gene. This could result in biologically counter-intuitive (though not impossible) situations. For example, the same dynamic histone mark "M" may indicate increased activation by binding sites of certain activator TFs and decreased activation by binding sites of other activator TFs. Similarly, the learnt parameters may be such that the same TF's binding site when coinciding with an appearance of an activating histone mark indicates upregulation but when overlapping with appearance of a different activating histone mark indicates downregulation. We therefore modified the model to rule out the above scenarios. In particular, rather than assign a free parameter $w_{T,M}$ to each T, M combination, we assigned to each TF "T" a free parameter $w_T$ and to each histone mark change "M" a free parameter $w_M$ and required that the parameter $w_{T,M}$ in the original model be equal to the product of $w_T$ and $w_M$. This requirement reduces the number of parameters drastically, from $\sim 160$ to $\sim 30$, affording us a far more constrained parameter estimation problem. Furthermore, it imposes the requirement that each TF has a fixed role (activator or repressor) and each histone mark change also provides evidence of a fixed regulatory change (activation or repression). Interpretations of the different combinations of signs of these two parameters are outlined in Additional file 4: Table S2.

Upon training the fw-pGENMi model on the entire dataset (in the up- and down-analysis modes separately), we found that the TF weights were largely consistent in directionality between the down-analysis and up-analysis (Fig. 2d), even though the model was trained independently for these two analyses. To us, this provided evidence of stability of the learnt model and reliability of the roles it learns for TFs and histone marks. We noted that most TFs, especially those with the greatest contributions to the models (e.g., top ranked TFs from Fig. 2b, c), had a positive weight (also see Additional file 1: Figure S9). This means that if a binding site for a TF overlaps with an activating histone mark such as H3K27ac, then the direction of change (increase or decrease) of the

histone mark is concordant with the direction of change of gene expression. In other words, most TFs exert an activating influence and the cellular state that has the activating mark present at the TFBS exhibits the higher gene expression. Figure 2e shows that the histone mark changes are consistent with the gene expression changes. For example, an H3K27ac mark that disappears in the late stage ("K27ac-down") has a positive weight in the down-analysis. This means that a disappearance of such a mark at an activator TF's binding site was indicative of downregulation. The same epigenomic change was assigned a negative weight in the up-analysis (i.e., disappearance of H3K27ac at an activator TF's site is evidence *against* upregulation of the associated gene). Both observations are consistent with our expectation for an activating histone mark. On the other hand, "K27ac-up" (an H3K27ac mark that appears in the late stage) was assigned a positive weight in the up-analysis and a negative weight in the down-analysis, again consistent with the biological expectations of an activating histone mark. Contrary to "K27ac-down," the histone mark change "K27me3-down" had a positive weight in the up-analysis, implying that such a change at an activator TF's binding site acts as evidence of upregulation of the associated gene; this is consistent with the repressive role reported in the literature for this mark [12]. The weights learnt for changes in the other two marks—H3K4me1 and H3K4me3—follow the same pattern as those of H3K27ac, consistent with their previously reported activating roles [12–14]. In summary, training of the fw-pGENMi model reveals the roles of TFs and histone marks involved in the down- as well as upregulation of genes in invasiveness.

### Epigenomic information improves model

Our models revealed the identities and roles of TFs underlying CRC progression by utilizing epigenomic profiles from different stages of invasiveness and combining those data with ENCODE ChIP-seq profiles of TFs. We next investigated the value of this strategy by contrasting its results with those from alternative strategies used within the same modeling framework. In particular, we compared the above strategy, henceforth called the "DiffMark" (differential histone mark overlapping with TF ChIP peaks), with (a) the use of TF DNA-binding (ChIP-seq) data alone ("TFBS-only" strategy), (b) the use of accessibility (ATAC-seq) profiles ("DiffAcc" and "PresAcc") in place of histone marks, (c) the use of changes in any histone mark in both directions ("DiffMarkAggr"), and (d) the use of presence rather than changes in histone marks ("PresMark"). Detailed descriptions of these strategies are presented in the "Methods" section.

The baseline strategy of using TF ChIP-seq data alone (TFBS-only) involves training pGENMi with one evidence type per TF, as opposed to the eight evidence types per TF used in the DiffMark strategy. The presence of a TF's ChIP peak within a certain maximum distance from the gene is treated as evidence of its potential regulatory influence on the gene. The "DiffAcc" (differential accessibility overlapping TFBS) strategy is similar to the DiffMark strategy but uses ATAC-seq peaks in early and late stages instead of the four histone marks. Thus, the pGENMi model for this strategy has two evidence types for each TF, one for either direction of change in DNA accessibility at the TFBS. For reasons mentioned below, we also tested a variant of this strategy ("PresAcc"—presence of accessibility at TFBS) where the presence of an ATAC-seq peak in early or late stages (or both), overlapping a TFBS, was considered as evidence of the TF's regulatory influence; the

corresponding pGENMi runs thus had only one evidence type per TF. In the "DiffMarkAggr" (differential histone marks aggregated for each direction separately) strategy, each evidence type for a TF represents whether the TF's binding site overlaps any histone mark change (in each direction) rather than a specific mark change; thus, this strategy utilizes two evidence types per TF in pGENMi modeling. The "PresMark" strategy (presence of histone marks) is similar to PresAcc, with one evidence type per histone mark indicating a TFBS that is flagged by that mark in either stage.

We relied on cross-validation to compare the above modeling strategies, all of which involve training pGENMi models with different definitions of *cis*-regulatory evidences. We partitioned all genes into training, validation, and test sets in proportions of 72%, 18%, and 10%, respectively; trained pGENMi parameters on the training set; used the validation set to pick optimal values for the two hyperparameters (including the distance threshold that defines gene regulatory regions); and computed the log likelihood ratio (LLR) of the model and a null model on the test set of genes. The process was repeated 100 times, with different random partitions, and the distribution of LLR scores of the model is shown in Fig. 3a, using the DiffMark strategy. There are two distributions shown, corresponding to down-analysis and up-analysis. The same evaluations were performed using each of the alternative strategies noted above, and the

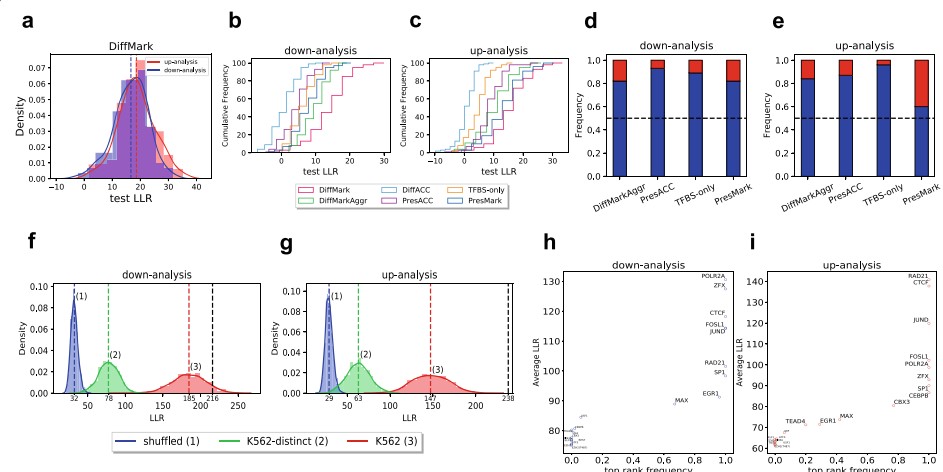

**Fig. 3** Comparison of the alternative strategies for defining *cis*-regulatory evidence. **a** Histogram of test LLRs derived from training-validation-test cross-validation for up- and down-analysis using DiffMark strategy. **b**, **c** Empirical CDFs of test LLR for different strategies (shown in different colors) suggest that DiffMark strategy performs better than alternatives in down- (**b**) as well as up-analysis (**c**). **d**, **e** Frequency of training-validation-test partitions (out of 100) where DiffMark results in a greater (blue) or lower (red) test LLR than an alternative strategy. **f**, **g** Histograms of LLR values on the entire dataset for three schemes designed to test the explanatory power of *cis*-regulatory evidences based on TF ChIP-seq data from a CRC (HCT116) cell line. Results are for down- (**f**) and up-analysis (**g**) using the DiffMark strategy. In "shuffled" scheme (blue), the model was trained using permuted evidence. In "K562" (red), ChIP-seq data from the K562 cell line, representing binding profiles of 20 randomly selected TFs, were used to generate the DiffMark evidence. The "K562-distinct" scheme (green) is similar to "K562," except that the 20 TF profiles were randomly selected from the 90 ChIP-seq profiles most dissimilar to the 20 HCT116 TFs. Colored dashed lines represent means of respective distributions. The LLR of the analysis performed using the 20 available ChIP-seq profiles from the CRC cell line (black dashed line) is significantly larger than the average of "shuffled" and "K562-distinct" schemes in both down- and up-analyses. **h**, **i** Statistical assessment of the contribution of each CRC TF in down- (**h**) and up-analysis (**i**), respectively. Each point represents one TF. The *y*-axis represents the average LLR of 100 models, each trained using the TF and 19 randomly selected K562-distinct TFs. The *x*-axis represents the frequency with which the TF is ranked as the most significant contributor among the 20 TFs in these models

corresponding distributions of test LLRs are shown in Additional file 1: Figure S10. Optimal distance thresholds were utilized for each strategy, and the distance dependence of accuracy for each strategy is shown in Additional file 1: Figure S7. We were surprised to note that the DiffAcc strategy, which relies on the changes in accessibility at a TFBS as *cis*-regulatory evidence, yielded test LLRs that were poorer than completely ignoring accessibility information (see the "Discussion" section). The PresAcc strategy yielded improved results, and our subsequent analyses therefore report on this method of utilizing accessibility data, rather than the DiffAcc method.

The cumulative distributions (CDFs) of all strategies are shown, for down- and up-analysis separately, in Fig. 3b, c, which reveals that the DiffMark strategy yields the highest test LLRs. The second-best LLRs were noted from the DiffMarkAggr strategy (for down-analysis) and the PresMark strategy (for up-analysis), both of which use histone marks (at TFBS) as *cis*-regulatory evidence but either ignore the specific identity of that mark (DiffMarkAggr) or ignore changes in marks (PresMark). All three strategies based on histone marks exhibit greater test LLRs than the TFBS-only strategy, which does not use any filter on TF ChIP peaks, and the PresAcc strategy, which uses only one type of epigenomic information (DNA accessibility) rather than four (histone marks). The PresAcc strategy improves upon the baseline TFBS-only strategy for up-analysis but has no effect for down-analysis. As a direct comparison of these two strategies, we asked if the DiffMark test LLR is greater than that of an alternative strategy on the same test set using 100 iterations of training, validation, and testing. The DiffMark strategy yields better test LLRs, indicative of an improved ability to "predict" expression based on unseen genes, in the vast majority of these head-to-head comparisons (Fig. 3d, e; Additional file 1: Figure S11). These results clearly demonstrate the value of utilizing histone mark changes as a filter on TFBS. The use of histone mark identities is noted to be valuable, as is the change in histone marks, especially for the down-analysis. Since genome-wide DNA accessibility profiles are known to correlate strongly with active enhancer marks such as H3K27ac, it was worth asking if a single accessibility profile is as informative as the four histone mark profiles; our results clearly indicate that this is not the case within the parameters of our comparisons.

Additional file 1: Figure S12 shows the ranking of TFs according to each of the alternative strategies, computed in the same manner as in Fig. 2b, c for the DiffMark strategy. Substantial agreement was noted among some strategies, in terms of the TFs that were utilized most for modeling differential expression. For example, the top six TFs in the down-analysis, as well as those in up-analysis, are identical between DiffMark and DiffMarkAggr. There were some major differences as well. For instance, JunD, which was ranked 3 and 4 respectively for down- and up- analysis by the DiffMark strategy, was not particularly informative for the TFBS-only strategy, which ranked this TF at 9 and 17 (out of 20 TFs) for down- and up-analyses, respectively. A possible explanation for this is that functional JunD sites have a relatively high tendency to be located distally from target genes, as the TFBS-only strategy relied on a distance threshold of 10 kbp for optimal performance. Conversely, the TF ELF1 was ranked relatively highly by the TFBS-only strategy (ranks 6 and 7) but was not found informative in the DiffMark strategy (ranks 15 and 12). Therefore, even though the baseline strategy shows poorer overall predictive ability, it may reveal complementary findings about important TFs.

## Model findings are specific to CRC cell lines

In the analyses so far, *cis*-regulatory evidences were based on TF ChIP-seq profiles for 20 TFs in a CRC cell line from the ENCODE Project. We next examined the significance of these TF profiles for the analysis. First, we established a random baseline where all *cis*-regulatory evidences were permuted (i.e., all evidences of the same type were randomly reassigned among genes). The model was then trained on the entire dataset (all genes) and LLR computed. Repeating this 1000 times, each time with a different permutation of evidences, we obtained a null distribution of LLR scores for down- and up-analysis (Fig. 3f, g; "shuffled"). The LLR scores obtained with the original (unpermuted) evidences, 216 and 238, respectively (Fig. 3f, g; "CRC"), were clearly far larger than those in the null distributions (means of 32.2 and 28.8, and standard deviations of 4.6 and 4.2, for down- and up-analysis, respectively). We next asked if the TF ChIP-seq profiles from a CRC cell line were more useful for the analysis than ChIP-seq profiles from a different cell line. For this, we repeated the analysis using 20 randomly selected genome-wide binding profiles for the myelogenous leukemia cell line K562 from the ENCODE Project. Using 1000 random selections of 20 out of 216 TFs, we obtained a distribution representing the accuracy of models that rely on real binding profiles from a different cell line (Fig. 3f, g; "K562"). The LLR scores obtained with the CRC ChIP-seq profiles were significantly greater than the mean of this distribution for the up-analysis, but not significantly larger in the down-analysis. This was surprising, since it suggests that epigenomic data might have similar explanatory power regardless of the associated cancer type. We noted that many of the K562 ChIP-seq profiles were highly similar to the CRC ChIP-seq profiles and, therefore, likely represented the same or related TFs (Additional file 1: Figure S13). The analysis was repeated using 20 TF ChIP-seq profiles from the K562 cell line that were randomly selected from a pool comprising 90 profiles that were most dissimilar to CRC profiles. This resulted in a distribution of LLR scores (Fig. 3f, g; "K562-distinct") that is closer to the null distribution and reveals the CRC profile-based LLR score to be highly significant—10.6 and 13.2 standard deviations above mean of the distribution (from K562-distinct profiles) for down- and up-analyses, respectively.

While the above analyses examined the significance of the entire set of 20 TF profiles from a CRC cell line, we also sought to quantify the significance of each of those TF profiles individually. To test the significance of the contribution of a particular TF's CRC ChIP-seq profile, we assessed the LLR score of a model based on 20 profiles consisting of the TF profile of interest and 19 randomly selected K562-distinct ChIP-seq profiles. We repeated this procedure 100 times and counted the frequency with which the CRC TF profile was ranked at the top of the 20 TFs. This was referred to as the "top-rank frequency" of the TF. A top-rank frequency close to 1 suggests that whenever this TF profile is used along with other profiles to model gene expression data in the CRC cell line, it tends to contribute the most to model performance. Figures 3h and i show this measure of each TF's significance for down- and up-analysis, respectively, and the average LLR score of models that utilized that TF's CRC profile. We noted that 6 out of 7 TFs ranked near the top in Fig. 2b, c also have a top-rank frequency equal to or close to 1, providing us with an objective way to assess CRC-relevant TFs, beyond the rankings shown in Fig. 2.

### Experimental validation of JunD as a regulator of CRC invasiveness

Our analyses above revealed the TFs JunD and FOSL1 as being significantly related to the transcriptional changes seen across different stages of CRC progression. Both TFs were predicted to be activators, according to down- as well as up-analysis. These TFs are part of the AP-1 complex, which has been previously reported to regulate multiple processes related to tumor invasiveness in a variety of cancers, including colorectal [32–34]. Here, we experimentally tested the role of one of these predicted TFs, JunD, in CRC migration and invasiveness. Knockdown of JunD (Fig. 4a) impaired both migration (Fig. 4b) and invasion (Fig. 4c) of SW480 cells. It is noted that SW480 cells display low levels of invasiveness (Additional file 1: Figure S1). Therefore, to determine the effects of disrupting JunD expression in a highly invasive cell model, JunD knockdown was repeated in the two SW480-M6 lines. In both lines, decreased expression of JunD caused marked reductions in migration and invasion (Additional file 1: Figure S14). Proliferation was not appreciably affected by disruption of JunD in either M0 or M6 cells (Additional file 1: Figure S15). Because the expression of AP-1 components can be auto-regulated by other AP-1 factors (e.g., [35]), we assessed the expression of various AP-1 genes following JunD knockdown. Using a $q$-value cutoff of 0.1, only FOSL1 and JunD were significantly downregulated (Additional file 4: Table S3). Collectively, these findings suggest that AP-1-mediated transcriptional activation of target genes may be integral to the invasion process in colorectal cancer.

### Characterization of TF "regulons" underlying CRC progression

In addition to identifying regulators of CRC invasiveness, our model also gives us the opportunity to characterize the genes that are potentially regulated by each TF and may thus act as mediators of a TF's regulatory influence on CRC invasiveness. To identify candidate target genes, we first examined the posterior probability of the hidden variable $Z_g$ (for each gene $g$) being equal to one. This event indicates that the model considers the gene as mediating the influence of one or more TFs on the transcriptomic changes between stages. We thus computed the posterior odds ratio ("POR") $\Pr(Z_g = 1|\ \text{data})/\Pr(Z_g = 0|\ \text{data})$ for each gene under the trained model. We then removed all *cis*-regulatory evidences corresponding to a particular TF ("T"), from the inputs to the model and recomputed the POR. We used the ratio of these two PORs (the full model and without the TF T), called ratio of posterior odds ratios ("RPOR"), as a measure of the TF's regulatory influence on that gene. The genes with the highest RPOR scores for each TF provide us a computational prediction of that TF's target genes or "regulon."

To characterize JunD targets experimentally, SW480-M0 cells were transduced with lentiviral particles expressing shRNA against JunD or a scramble control shRNA. RNA-seq identified 2133 genes that are differentially expressed ($p$-value < 0.05) upon JunD knockdown. This set, henceforth called "JunD-KD-DE," is expected to include both direct targets of JunD as well as indirect targets. The differentially expressed genes were significantly enriched (hypergeometric test $p$-value 0.002) for the genes that were differentially expressed between M0 and M6 cells and also model-predicted to be JunD targets (top 500 RPOR scores in down-analysis), further supporting a central role for JunD in driving the transcriptomic changes associated with CRC progression. We

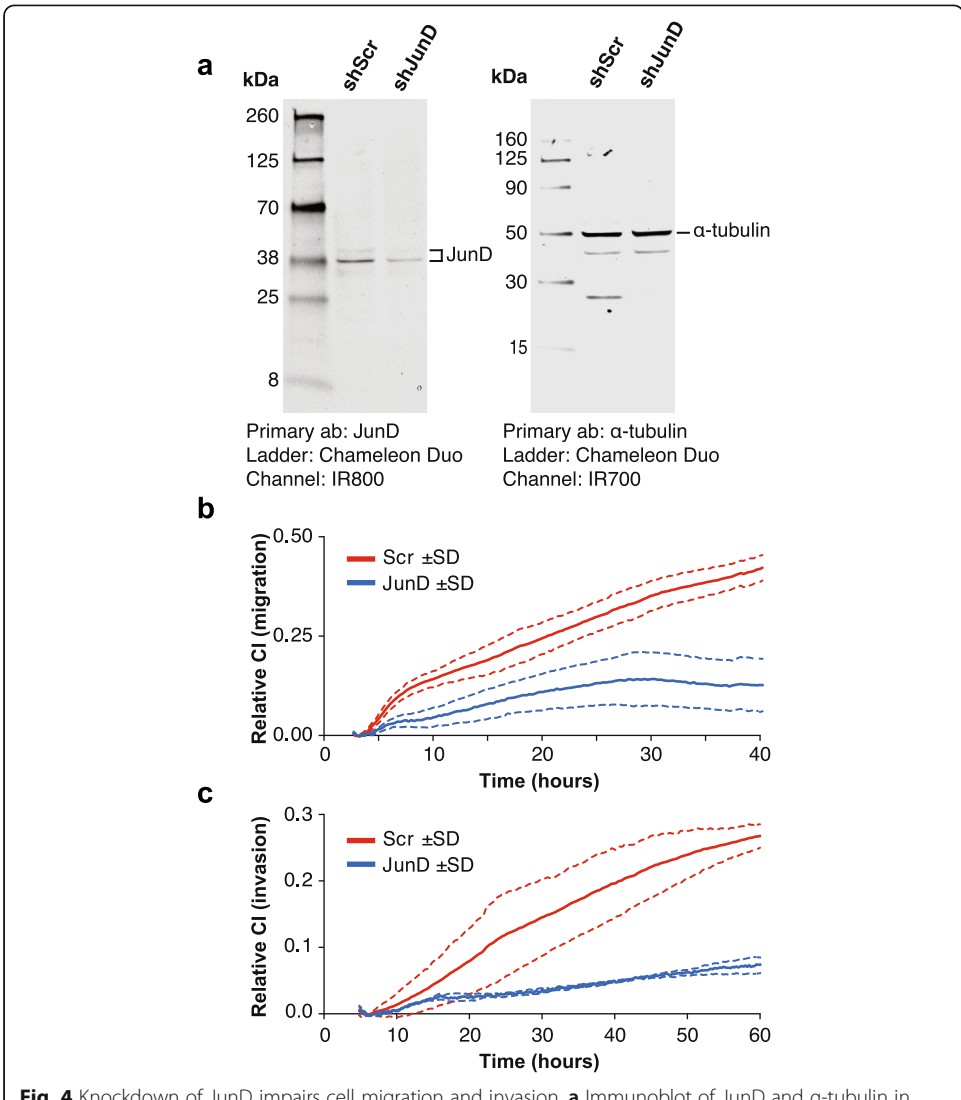

**Fig. 4** Knockdown of JunD impairs cell migration and invasion. **a** Immunoblot of JunD and α-tubulin in lysates from SW480 (M0) cells expressing JunD shRNA knockdown (shJunD) or scramble control (shScr). **b** Migration for JunD knockdown (blue line) and scramble control (Scr; red line) was monitored continuously over 40 h using a xCelligence realtime cell analysis platform with cell invasion migration (CIM) plates. Fetal bovine serum was used as a chemoattractant. Cell index (arbitrary units) corresponds to cell migration capacity. Dotted lines represent the standard deviation (SD) of three independent cultures measured in parallel. **c** Cell invasiveness was measured continuously for 60 h using CIM plates that were precoated with Matrigel. All other parameters are the same as for **b**

subsequently tested if the JunD-KD-DE gene set was enriched in the model-predicted mediators of JunD. As shown in Fig. 5b, for varying thresholds for defining predicted mediators, the enrichment was statistically significant at a nominal *p*-value threshold of 0.05. The overlap between the top 500 predicted mediators and the JunD-KD-DE set was significant with a *p*-value of 3E−5 (Fig. 5a). Notably, the top 10 predicted mediators included 5 genes that were DE upon JunD knockdown: SERINC2, COBL, FGF3, KALRN, and CLDN4. This suggests that our computational procedure correctly identified JunD targets through *cis*-regulatory evidences based on its ChIP-peaks and dynamic epigenomic information.

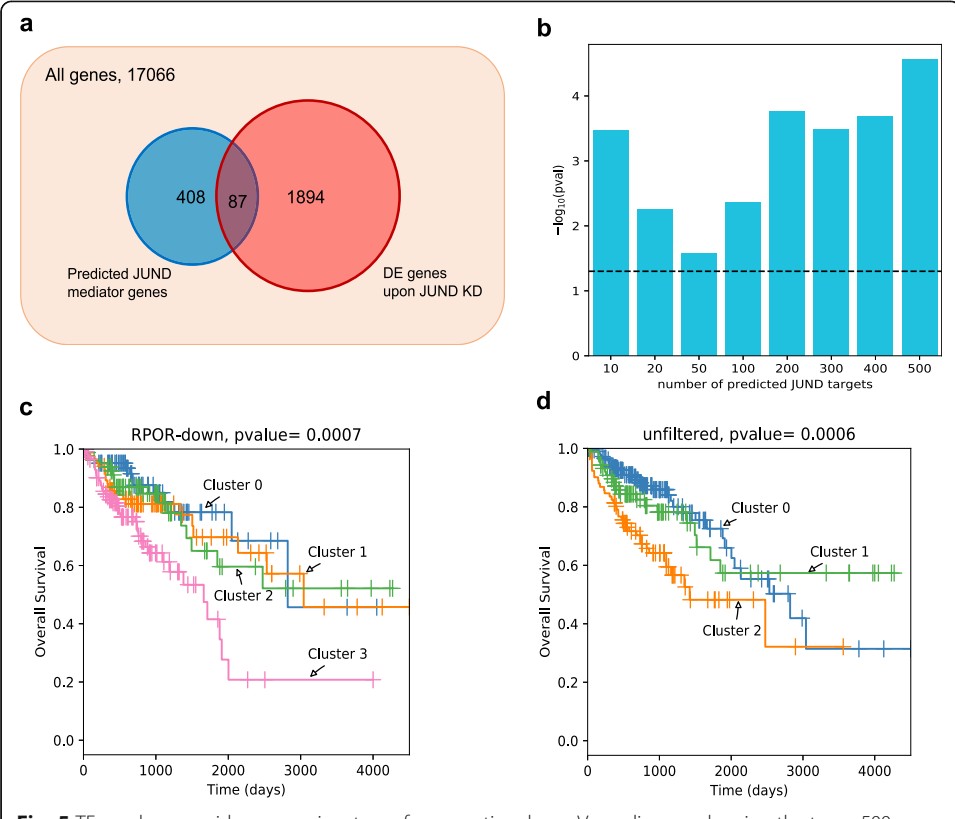

**Fig. 5** TF regulons provide a gene signature of prognostic value. **a** Venn diagram showing the top ~ 500 predicted target genes with the highest RPOR in the down-analysis and 1981 JUND-KD-DE genes with *p*-value < 0.05. Their intersection (87 genes) has a hypergeometric test *p*-value of 3E–5. **b** Significance (-log10 of hypergeometric test *p*-values) of the overlap between the JUND-KD-DE gene set and predicted target gene sets of varying sizes. The target genes were predicted based on their RPOR value in the down-analysis. The dashed horizontal line represents a *p*-value of 0.05. **c**, **d** The Kaplan-Meier survival analysis performed on the overall survival of 374 COAD patients from TCGA. In **c**, patients are clustered into four groups using expression profiles of the top 70 genes with the highest RPOR in our down-analysis, as well as somatic mutations associated with these genes and the expression of miRNAs targeting them. In **d**, patients are clustered into three groups using the expression profiles of all genes, as well as with somatic mutations associated with them and the expression profiles of miRNAs targeting the genes

A subset of genes that were computationally predicted to be mediators of JunD influence on CRC progression is tabulated in (Additional file 5: Tables 1, 2). These genes were differentially expressed upon knockdown of JunD and were differentially expressed between stages. Notably, these predicted mediators include a number of genes that are individual prognostic markers for CRC survival and factors that have been implicated in tumor metastasis, whether in colorectal cancer or in other solid tumor types. For example, ELF3 [36, 37], ITGA6 [38, 39], DLG3 [40], LRP11 [41], and TPBG [42] have all been linked to the WNT/β-catenin signaling pathway. Dysregulation of WNT signaling is a hallmark of colorectal cancer that controls processes relating to cancer development and progression [43]. Some dysregulated genes provide possible links between various cancer-associated pathways. In addition to promoting proliferation mediated by WNT signaling, ITGA6 is regulated by MYC in CRC [44], suggesting that the gene may be part of a feed-forward mechanism between MYC and WNT signaling [39]. GDF15 is a member of the TGF-β superfamily that has been shown to promote CRC metastasis in vitro and in vivo through activation of the

Smad2/3 pathway via binding to the TGF-β receptor [45]. GDF15 has been shown to be activated c-Fos [46], an AP-1 component, which is consistent with the altered expression noted in JunD knockdown cells. Another gene involved in TGF-β signaling [47], MGAT5, has been linked to tumor growth [48, 49] and invasion [50, 51], as well as maintenance of colon cancer stem cells [49]. The identification of genes linked to known pathways related to CRC development and progression lends further validity to the analytical approach.

In light of the promising results above indicating our ability to identify TF targets that are coordinately regulated (regulons), we extended the approach to identify the highest confidence mediators of 7 TFs' influence on CRC invasiveness (union of top 5 TFs in up- and down-analyses), thus constructing a "gene signature" of the phenotype. We used this signature to cluster multi-omics profiles (gene expression, miRNA expression, and somatic mutations) of colorectal cancer patients from the TCGA database [52], and performed survival analysis on the resulting clusters (see the "Methods" section). The clustering was performed after limiting the omics profiles to genes in the signature, using the network-guided clustering pipeline in the KnowEnG system [21] for clustering somatic mutation profiles. We performed the analysis using different numbers of clusters (3 or 4), size of signature (50, 70, or 100 genes), degrees of influence of the network (smoothing factor of 0.3 or 0.8), and network types (protein-protein interaction network or pathway co-membership network). Survival analysis using the best of these parameters (Fig. 5c) showed statistically significant difference in survival characteristics of patient clusters (log-rank test *p*-value 0.0007, adjusted *p*-value 0.03 after correcting for multiple modes of clustering). This result was comparable to the best clustering obtained using the complete dataset (i.e., no filtering of genes) (Fig. 5d, Additional file 4: Table S4, *p*-value 0.0006), indicating that the gene signature was able to capture the survival-related information present in the complete multi-omics profiles in only a small subset of genes. Moreover, survival analysis using the model-based gene signature yielded better stratification of patients than when using the top genes based on differential expression between stages (Additional file 4: Table S4) (this latter analysis was also repeated with different settings of the clustering parameters, for a fair comparison to the above results).

## Discussion

We present here a comprehensive multi-omics approach to investigate the *cis*-regulatory mechanisms underlying the biological processes marked by large-scale transcriptomic changes. By adopting this approach to the study of CRC invasiveness in a well-controlled experimental setting, we identified the major regulators of this process as well as some of their key mediators. Our approach identified numerous TFs and downstream targets known to be involved in the metastasis-related process, as well as additional factors that have not yet been directly studied in the context of tumor progression (or in the context of colorectal cancer, specifically). Identifying these latent changes has the potential to greatly improve our understanding of the complex regulatory processes that control the metastatic progression and to help identify novel features of cancer that can be therapeutically targeted. While many cancers share common general features (e.g., alterations that affect conserved oncogene-related pathways), the specific pathway disruptions within each cancer are largely unique and

dynamic. This is true not only at the "cancer type" level, but also at the inter- and intra-individual levels.

Our approach is complementary to a large body of work that seeks to identify regulators based on correlations in transcript levels of TFs and their targets [18] ("*trans*" evidence), in some cases augmented with data on TF-DNA binding ("*cis*" evidence) [19]. The co-expression approach, including that of the popular WGCNA tool [53], is better suited to characterizing conserved changes that occur across a large numbers of samples for which expression profiles are available, e.g., from patient data in TCGA cohorts [54]. However, when investigating specific stages, subtypes, and/or features of a given cancer, sample numbers are usually restricted, which limits the suitability of those approaches. In our experimental paradigm, where we seek to characterize CRC invasiveness in a highly controlled setting using only a handful of biological samples, the correlation-based approach is far less practical. However, it is still possible to discern transcriptomic and epigenomic shifts with statistical significance, and our analysis exploits this source of information to connect TFs to their target genes. Our use of dynamic epigenomic evidence was crucial to the effectiveness of our approach, as we found by comparing its predictive ability to alternatives where the same model was trained with such evidence partially or completely removed. At the same time, we believe that further work is needed to fully understand the pros and cons of relying on changes in histone marks versus simply the presence of these marks. Another important feature of our analysis is its simple approach to account for directionality of regulatory influences. In assigning each TF and each dynamic histone mark a learnable weight (whose sign indicates activating/repressive influence), we made a simplification that allows us to learn the predominant value of that *cis*-regulatory evidence from the data. We observe the learned weights to be consistent between analyses, arguing for their reliability, and found them also to agree with previously reported interpretations of the histone marks.

One of the surprising findings from our comparisons of different schemes was the poor performance of the DiffAcc scheme, where a TFBS is considered functional if it overlaps an ATAC-seq peak exclusive to one of the stages. This scheme is thus similar to the DiffMark scheme but relies on accessibility profiles instead of histone mark profiles. Its cross-validation performance was poor compared not only to DiffMark but also the TFBS-only scheme, which completely ignores accessibility data. It is possible that our implementation of this scheme requires a more careful definition of differential accessibility, and future work will reveal the strength of this approach. Indeed, our alternative strategy for utilizing accessibility profiles—the PresAcc scheme—did exhibit better predictive ability than TFBS-only in the up-analysis. The relative failure of the DiffAcc scheme may also have a biological reason: accessibility changes during CRC progression may be more quantitative, and therefore, inadequate information is captured when represented as a binary change, compared to histone mark changes. As such, our approach of using accessibility events as a binary variable could err in the trade-off between sensitivity and specificity.

We note that the analysis framework in our work bears superficial resemblance to a classification setting, where the *cis*-regulatory evidences associated with a gene may be used to predict the differential regulation of that gene. This would require a hard threshold on the differential expression *p*-value, and demand that every designated DE

gene bears one or more *cis*-regulatory evidences. Our probabilistic approach avoids the use of hard thresholds on differential expression and retains the information about DE strength as reflected in the *p*-value. Furthermore, it allows for the possibility that many DE genes may lack *cis*-regulatory evidences, perhaps due to the limitations of our ability to recognize those evidences. Another technical point worth noting is that our modeling was performed separately for upregulation and downregulation between stages. The pGENMi model attempts to explain the *p*-values of differential expression, but these *p*-values do not contain information on the directionality of the expression changes. As such, in a "combined" analysis, where both types of differential expression *p*-values are present, the model attempts to use the same *cis*-regulatory evidence, with the same weight, to explain both up- and downregulation and will be confounded; the resulting weights will be less reliable. This is the primary reason why we separated the up- and down-analyses.

The analytical approach we describe herein offers several opportunities for expanded mechanistic and computational studies. Firstly, our analysis identified TFs known to play an important role in CRC invasiveness, as well as additional TFs with a less-defined contribution. Our method does not directly predict the overall direction of influence, however. For example, our approach is unable to predict whether the knockdown of JunD would be expected to increase or decrease cellular invasiveness, only that JunD disruption would likely alter the phenotype. Future work may address this by explicitly modeling phenotypic difference as a function of gene expression changes, which in turn are related to regulator influences. Such multi-level networks of influence (from TFs to genes to phenotype) will be an important frontier of research. A second direction of improvement will be to incorporate *cis*-regulatory evidences in a non-binary manner. For simplicity of modeling, we currently encode each combination of TF and histone mark through a binary evidence per gene; the strengths and multiplicity of such evidence may be rigorously accounted for in future models, perhaps borrowing from previous work in the context of TFBS analysis [55]. Finally, it will be exciting to integrate the multi-omics data and analysis presented here with information about TF-gene co-expression from patient cohort studies such as those in the TCGA [54] and clinical trials.

## Methods

### Cell line generation

SW480 (CCL-228) cells were obtained from the American Type Culture Collection (ATCC, Manassas, VA) and maintained at 37 °C in a humidified incubator with 5% $CO_2$ atmosphere. Cells were cultured using Dulbecco's modified Eagle's medium (Corning Life Sciences, Corning, NY) supplemented with 10% FBS (MilliporeSigma, Burlington, MA), 100 U/mL penicillin (Corning), and 100 μg/mL streptomycin (Corning). For selection of invasive sub-populations, SW480 cells were serum-starved in FBS-free culture media for 16 h and released from the culture plate surface using 0.05% Trypsin (Corning). Two million cells were plated in FBS-free culture media in 8.0-μm permeable transwell supports (Corning) that had been coated with Matrigel Growth Factor Reduced (GFR) Basement Membrane Matrix (Corning) and set in 6-well plates containing media supplemented with 10% FBS as a chemoattractant. Cells were allowed

to invade for 24 h. Invaded cells were harvested from the underside of supports using Trypsin. Cultures were expanded, and the invasion process was repeated. Cell lines were periodically monitored for mycoplasma using Hoechst staining (MilliporeSigma). Culture health and identity were monitored by microscopy and by comparing population doubling times to baseline values recorded at time of receipt. Additional authentication of this cell line above that described was not performed.

### Knockdown of JunD

To disrupt JunD expression, shRNAs from The RNAi Consortium (TRC) were screened for knockown potential in SW480 cells by transient transfection of purified plasmid using Mirus TransIT-LT1 (Mirus Bio, Madison, WI) followed by western blotting to monitor JunD expression (MAB5526 antibody; R&D Systems, Minneapolis, MN). TRC clone TRCN0000014975 was found to reduce JunD expression to the greatest extent (data not shown) and was used in subsequent experiments. Lentiviral plasmids were co-transfected with packaging vectors psPAX2 (Addgene plasmid #12260; a gift from Didier Trono) and pMD2.G (Addgene plasmid #12259; a gift from Didier Trono) into HEK293T/c17 cells (ATCC) using Mirus TransIT-Lenti. Virus-containing medium was collected 48 h after transfection and cleared of potential cells using 0.45-μm Steriflip filters (MilliporeSigma). Virus-containing media were mixed with Polybrene (MilliporeSigma) for transduction. Expression of JunD was assessed using the MAB5526 antibody. Scramble control shRNA on the same vector backbone (pLKO.1-puro non-target control shRNA; MilliporeSigma) was used as a control.

### Measurements of cell migration, invasion, and proliferation

Cell migration and invasion were assessed using an xCELLigence Real Time Cellular Analysis (RTCA) DP instrument (Acea Biosciences, San Diego, CA). Migration was measured using uncoated CIM-Plate 16 (Acea) plates. For invasion assessments, plates were pre-coated with a 1:20 dilution of Matrigel GFR (Corning). To monitor invasion and migration, cells were serum-starved for 16 h, collected by trypsinization, and plated at 200,000 viable cells per well in the top chamber of a CIM-Plate 16. Viable cell counting was performed using propidium iodide staining with quantitation on an Acea NovoCyte 3000 RYB flow cytometer. Ten percent of FBS-containing media was used as a chemoattractant in the bottom chamber. Cells were allowed to settle for 10 min at room temperature prior to loading CIM-Plates onto the xCELLigence DP. Impedance data were acquired at 15-min intervals for 40 h (migration) and 60 h (invasion). Proliferation was assessed using an xCELLigence RTCA MP instrument (Acea). Transduced cells were plated at 20,000 viable cells per well in E-Plate View 96 plates, allowed to settle for 10 min at room temperature, and loaded onto the xCELLigence MP, and impedance data acquired at 15-min intervals for 60 h.

### Transcriptome sequencing (RNA-seq)

Total RNA was extracted using the Direct-zol RNA Kit (Zymo Research, Tustin, CA). For the assessment of parental SW480 cells and selected invasive lines, TruSeq (Illumina) libraries were prepared, and paired-end 150 base pair sequencing was performed on an Illumina HiSeq 4000 (San Diego, CA) in the Mayo Clinic Medical Genome

Facility. For studies related to JunD knockdown, TruSeq Stranded mRNA libraries were prepared and sequenced on an Illumina NovoSeq 6000 using a 150 PE flow cell at the University of Minnesota Genomics Center.

### Differential expression analysis on RNAseq data

Adapter sequences were removed using the TrimGalore wrapper around Cutadapt [56], and reads were aligned to the human genome (hg19) using HISAT2 [57]. Transcript assembly and quantification were performed using StringTie [58]. For DE analysis, Ballgown [59] was used to calculate log2 fold changes, *p*-values, and false discovery rates (FDR).

### Chromatin immunoprecipitations

Five million actively growing cells were collected, suspended in PBS, and cross-linked using 1% formaldehyde (final concentration). Cross-linking was quenched using 125 mM glycine at room temperature, followed by two washes using PBS. Cells were pelleted and resuspended in cold lysis buffer consisting of 1% Triton-X, 0.1% sodium deoxycholate, proteinase inhibitor cocktail (MilliporeSigma), and Tris-EDTA solution. Lysates were incubated on ice for 10 min, diluted with TE, sonicated for 15 min (30 s on/30 s off) using a Bioruptor Pico (Diagenode, Denville, NJ), and cleared using centrifugation. Supernatants were transferred to a fresh tube, and DNA content was determined using Qubit fluorometric quantification (Thermo Fisher Scientific, Waltham, MA). Chromatin was incubated with relevant antibodies and isolated using protein G couple magnetic beads (Thermo Fisher). Beads were washed with a buffer consisting of 50 mM Tris-HCl, 10 mM EDTA, 100 mM NaCl, 1% Triton X-100, 0.1% sodium deoxycholate at pH = 8.1, followed by a high salt buffer containing 500 mM NaCl (all other components remained the same) and LiCl buffer (10 mM Tris-HCl, 0.25 M LiCl$_2$, 0.5% NP-40, 0.5% sodium deoxycholate, 1 mM EDTA, pH = 8.0). Bound chromatin was eluted and crosslinking reversed. DNA was treated with RNase A and proteinase K before being purified using the Qiagen MinElute PCR Purification Kit (Valencia, CA). ChIP-seq libraries were prepared from ChIP DNA using the NEBNext Ultra II DNA Library Prep Kit (New England Biolabs, Ipswich, MA). Libraries were sequenced to 51 base pairs using paired-end mode on an Illumina HiSeq 4000 (San Diego, CA) in the Mayo Clinic Medical Genome Facility.

### ChIP antibodies

Antibodies used for ChIP consisted of anti-H3K27ac (8173; Cell Signaling Technology, Danvers, MA), anti-H3K4me1 (ab8895; Abcam, Cambridge, MA), anti-H3K4me3 (purified antibody generated in-house by the Mayo Clinic Epigenomics Development Lab, Rochester, MN [60], and anti-H3K27me3 (9733, Cell Signaling Technology).

### ChIP-seq data analysis

Sequences were aligned to the human genome (hg19) using Bowtie2 [61]. Peak calling for H3K27ac, H3K4me1, and H3K4me3 signals was performed using MACS2 [62]. SICER [63], which was used to call peaks for H3K27me3 data. FDR thresholds of 0.01 were used for all peak calling.

### ATAC-seq

ATAC-seq library construction was performed as previously described [64]. Fifty thousand cells were lysed in cold ATAC-Resuspension Buffer (RSB) containing 0.1% NP40, 0.1% Tween 20, and 0.01% digitonin on ice. Lysis buffer was washed out with cold ATAC-RSB containing 0.1% Tween 20 followed by centrifugation at 4 °C. Nuclei-containing pellets were resuspended in transposition mix containing Tagment DNA buffer (Illumina), Tn5 Transposase (Lucigen, Middleton, WI), and 0.05% Tween 20. Reactions were incubated for 30 min at 37 °C with constant agitation. Transposed DNA was purified using QIAgen MinElute columns. DNA was amplified using Nextera sequencing primers (Illumina) and NEB High Fidelity 2× PCR Master Mix (New England Biolabs) for 3–5 cycles. PCR-amplified DNA was purified using QIAgen MinElute columns and sequenced to 51 base pairs using paired-end mode on an Illumina HiSeq 4000 in the Mayo Clinic Medical Genome Facility. Adapter sequences were removed using the TrimGalore wrapper around Cutadapt [56], and reads were aligned to the human genome (hg19) using Bowtie2 [61]. Duplicate reads were removed using Picard Tools [65]. Peak calling was performed using MACS2 [62].

### Gene set characterization of DE genes

Up- and downregulated genes with adjusted $p$-value < 0.05 were separately analyzed for enrichment of gene sets from the mSigDB [22] collection (C2, C4, C6, C7) using the KnowEnG [21] platform's Gene Set Characterization pipeline without network guidance.

### TF-DNA binding profiles

TF ChIP-seq data for the colon cancer cell line HCT116 (20 TFs) were downloaded from the ENCODE Project web site (see Additional file 4: Table S1). These included all 20 TFs for which the available ChIP-seq profiles had high read depth. Data for the K562 cell line (216 TFs) were downloaded from the same source (Additional file 4: Table S5).

### pGENMi input generation

#### DiffMark

We first determined dynamic histone mark sites as histone modification ChIP-peaks (FDR 0.01) present in both replicates of either M0 or M6 profiles but without an overlapping ChIP-peak for the same modification in the other stage. These sites were grouped separately by the identity of histone modification and by the direction of change ("down"—present only in M0; "up"—present only in M6). The DiffMark evidence was then generated by intersecting the ENCODE TF binding sites from HCT116 cell line with dynamic histone mark sites, retaining information about the histone mark type and direction of change. The *cis*-regulatory evidence representing a TF T and dynamic histone mark M for gene $g$ was set to 1 if the binding site of T overlapped a dynamic histone mark of type M within distance $d$ upstream or downstream of the gene; if such evidence was present for histone mark in the up direction as well as the same mark in the down direction, only the direction associated with the largest change was considered. In different tests, the parameter $d$ was set to 10 kb, 50 kb, 200 kb, or 1 Mb.

### DiffMarkAggr

DiffMark evidence was used to generate the DiffMarkAggr evidence by computing the disjunction of the binary *cis*-regulatory evidences of all histone marks for each direction separately. This resulted in two evidence bits per TF, gene pair and a 40-dimensional evidence vector representing all 20 TFs.

### PresMark

The presence of ChIP peak (FDR 0.01) of a specific histone modification, in either stage, overlapping with a TF ChIP peak within the distance threshold, was encoded as "1" and zero otherwise. This resulted in four evidence bits per TF (one for each histone modification), gene pair and an 80-dimensional evidence vector representing all 20 TFs.

### DiffAcc and PresAcc

Using ATAC-seq profiles, DiffACC evidence was produced by a similar procedure as DiffMark, except that ATAC-seq peaks (FDR 0.05) were used in place of histone mark ChIP-seq peaks to intersect with TFBS. This gave us two evidence bits (one for either direction of accessibility change) per TF, gene pair, and a 40-dimensional evidence vector overall. In generating the PresACC evidence, the presence of an ATAC-seq peak in either stage, overlapping with a TF ChIP peak within the distance threshold, was encoded as "1" and zero otherwise. This resulted in a 20-dimensional evidence vector for each gene.

### TFBS-only

Finally, the TF ChIP profiles were used to generate TFBS-only evidence with the presence of TFBS for a gene encoded to one leading to a 20-dimensional evidence per gene.

### DE genes

We analyzed 17,200 protein-coding genes for which the annotations were downloaded from GENCODE [66]. To study the mechanisms leading to the upregulation and downregulation of genes separately, in the down- (resp. up-) analysis, we replaced DE *p*-values with $1 - p$-value for every gene with fold change greater (resp. less) than 1.

### pGENMi model

The pGENMi model was presented in [26], and we outline it here with terminology suitable for our context. The model uses a hidden binary variable $Z_g$ for each gene $g$, representing whether or not the gene's differential expression is associated with one or more of the TFs. Each *cis*-regulatory evidence contributes with a unique weight to the prior probability of this hidden variable as follows:

$$\Pr(Z_g = 1) = \frac{1}{1 + \exp\left(-\left(w_0 + \sum_t \sum_m w_{tm} r_{gtm}\right)\right)}$$

where $r_{gtm}$ represents a (binary) *cis*-regulatory evidence in the form of a binding site for TF $t$, flagged by dynamic histone mark $m$, in the regulatory region of gene $g$, and $w_{tm}$ is the weight associated with it (same for all genes).

The distribution of the DE $p$-value $P_g$ of a gene $g$ is conditioned on $Z_g$. If $Z_g = 1$, this $p$-value follows a beta distribution with trainable parameter $\alpha$, which specifies the skewness of the distribution toward the significant $p$-values; if $Z_g = 0$, $P_g$ is assumed to be uniformly distributed.

$$P_g = \begin{cases} \text{Unif}(0,1) \; \textit{if } Z_g = 0 \\ \beta(\alpha,1) \; \textit{if } Z_g = 1 \end{cases}$$

The model is trained by maximizing the regularized likelihood of the data, assuming the genes to be independent. $L_2$-regularization was used in the objective function, as shown below:

$$\mathcal{L}(\theta) = \log\left( \Pr_\theta\left( \overrightarrow{P_g} \right) \right) - \lambda \sum_t \sum_m w_{tm}^2$$

where $\overrightarrow{P_g}$ is the vector of DE $p$-values of all genes, $\theta$ is the set of all trainable parameters $\{ \overrightarrow{w}, \alpha \}$, and $\lambda$ is the regularization coefficient, a hyper-parameter of the model. The explanatory power of a model utilizing the given *cis*-regulatory evidences ($H_1$) is measured by the difference between its log-likelihood and that of a null model ($H_0$) that does not use *cis*-regulatory evidence:

$$LLR = log\left( \Pr_{\theta_1}\left( \overrightarrow{P_g} | H_1 \right) \right) - log\left( \Pr_{\theta_0}\left( \overrightarrow{P_g} | H_0 \right) \right)$$

The factorized-weights pGENMi (fw-pGENMi) model has the same formulation as pGENMi, except that the free parameter $w_{tm}$ is replaced by $w_t w_m$, where $w_t$ is a free parameter for TF $t$ and $w_m$ is a free parameter for dynamic histone mark $m$. With 20 TFs and four histone marks that can change in either of two directions, this model uses only 28 weights to aggregate the regulatory evidence, as opposed to pGENMi, which requires 160 weights. The model was trained using a distance threshold of 50 kb, derived from pGENMi cross-validation runs (Fig. 2a), and the regularization coefficient was retrained.

pGENMi and fw-pGENMi were implemented in PyTorch, and model optimization was performed using the Adam stochastic gradient descent variant [67].

### Dissimilarity criterion in "K562-distinct" scheme

To find the TF ChIP-seq profiles from K562 cell line that are most dissimilar to the CRC TF ChIP-seq profiles, first the DiffMark *cis*-regulatory evidence matrix was computed using ChIP-seq data of 216 TFs from K562. For each gene, the disjunction of all eight bits corresponding to a TF was used as a feature representing that TF's *cis*-regulatory evidence associated with the gene, and the TF was then represented by a feature vector, with one feature per gene. Next, the pairwise similarity between each K562 profile and CRC profile was calculated by the Jaccard similarity score between their corresponding feature vectors. For each K562 TF profile, the highest similarity score to a CRC TF profile was used as the "CRC similarity score" associated with the K562 TF, and all K562 TFs were ranked by their CRC similarity score. Some of the CRC TFs were also profiled for DNA-binding in K562 cell line, and we chose the minimum CRC similarity score ($\sim 0.2$) among these TFs to set the cutoff. The K562 TFs with CRC similarity scores less than 0.2 (90 TFs) were used in K562-distinct scheme.

### Gene signature for CRC invasiveness

To construct a gene signature, we considered the top 5 TFs from up- and down-analysis under the DiffMark strategy, which included a total of 7 TFs: JUND, FOSL1, CTCF, ZFX, RAD21, MAX, and POLR2A. For each gene, we calculated the product of the RPORs associated with these top TFs and used the top 50, 70, or 100 genes ranked by this product as the gene signature. The signature was then used to cluster multi-omics profiles of 374 COAD patients from TCGA. Multi-omics clustering was performed, using the KnowEnG system, on the gene expression and somatic mutations associated with the genes in the signature, and the expression of miRNAs targeting them (miRNA-gene interactions were obtained from miRTarBase release 7.0 [68]). First, the patients were clustered using each data type (network-guided clustering was used for somatic mutation profiles), then the clustering of the cluster assignments specified the final stratification of the patients. The entire procedure was repeated for the up-analysis (RPOR_up) and down-analysis (RPOR_down) separately (Additional file 4: Table S4). Kaplan-Meier survival analysis was performed on the clustered patients using the KnowEnG system, and the log-rank test *p*-values, indicating the difference in the survival times of the clusters, were used to choose the best gene signature. The *p*-values were subjected to Bonferroni correction for all tests performed using the same gene set (model-based signature, DE genes, or unfiltered).

## Supplementary Information

---

**Additional file 1.** Supplementary Figures.

**Additional file 2.** Gene set characterization. Gene set characterization of the DE genes between early and late stages. The downregulated genes are significantly enriched for cancer-related gene signatures from mSigDB.

**Additional file 3.** Annotated list of genes with SNVs that exhibited significant shifts in alternative allele frequency between M0 and M6.

**Additional file 4: Table S1.** List of HCT116 TFs and their corresponding files obtained from ENCODE portal (https://www.encodeproject.org/). **Table S2.** Interpretations of the different combinations of signs of TF and histone mark change weights. **Table S3.** Expression changes in AP1-complex components upon JunD Knockdown. **Table S4.** Survival analysis on TCGA COAD cohort. Survival analysis of 374 patients was performed using different gene sets for clustering of omics profiles. Each row indicates the criterion used to pick the gene set (gene set type) and the setting used in the best clustering of patients in terms of survival analysis *p*-value (pval). The gene set was chosen to be of one of the following: 1. Using the data of all genes provided (Unfiltered). 2. Genes with the highest RPOR value in down-analysis (RPOR_down). 3. Genes with the highest RPOR value in up-analysis (RPOR_up). 4. Significantly differentially expressed genes (DE Genes). For all types of gene sets except "Unfiltered" sizes of 50, 70, and 100 were tested. To find the best clustering for the patients using each gene set, two types of networks, viz., protein-protein interaction network (PPI) and Enrichr pathway network were tested using the knowledge-guided clustering pipeline in KnowEnG; two different settings of smoothing (0.3 and 0.8), representing different levels of network-guidance, were explored. (Knowledge-guided clustering was applied only to the sparse somatic mutation data, while other types of omics profiles were clustered in the default mode.). Cluster sizes of 3 and 4 were tested. The total number of clusterings explored and followed by survival analysis, for RPOR-down and RPOR-up combined, DE Genes, and Unfiltered, are 48, 24, and 8 respectively, which were used to adjust the *p*-values by Bonferroni Correction (adj. p-val). **Table S5.** List of K562 TFs and their corresponding files obtained from ENCODE portal (https://www.encodeproject.org/).

**Additional file 5.** Predicted mediators of JunD influence on CRC progression. **Table 1.** contains the model-predicted targets of JunD (top 500 RPOR scores in down-analysis) that were differentially expressed upon knockdown of JunD (p-value < 0.05), and were differentially expressed between stages (p-value < 0.05). **Table 2.** genes have the same characteristics as **Table 1** genes except for being predicted as JunD targets in up-analysis.

**Additional file 6.** Review History.

---

#### Acknowledgements
The authors wish to acknowledge Seid Hamzic, Ph.D., for the analytical support.

#### Review history
The review history is available as Additional file 6.

**Peer review information**

**Authors' contributions**
SG, SS, and SMO designed the study and wrote the manuscript. SG performed the data analyses. CH contributed to the data pre-processing. RES and KJB performed the laboratory experiments related to JunD knockdown studies. The authors read and approved the final manuscript.

**Funding**
This work was supported in part by the National Institutes of Health (grant R35GM131819 and U54-GM114838 to SS), the CompGen Initiative at UIUC (CompGen fellowship to SG), and the Mayo Clinic Center for Biomedical Discovery (Discovery Science Award to SMO). The funding agencies did not play any role in the design of the study; collection, analysis, and interpretation of the data; and writing of the manuscript.

**Availability of data and materials**
All datasets used and/or analyzed during the current study are available from the corresponding authors on request. Sequence data used for this study have been deposited to the NCBI Sequence Read Archive (SRA) and are accessible under BioProject number PRJNA659546 (https://www.ncbi.nlm.nih.gov/sra/?term=PRJNA659546) [69].
Tables of ENCODE accession numbers for HCT116 and K562 cell lines are provided as Additional file 4: Table S1 and Table S5, respectively. The TCGA COAD profiles were obtained from https://xenabrowser.net/datapages/?cohort=GDC%20TCGA%20Colon%20Cancer%20(COAD)&removeHub=https%3A%2F%2Fxena.treehouse.gi.ucsc.edu%3A443. The software developed here are available from https://github.com/sabagh1994/fw-pGENMi [70] and https://doi.org/10.5281/zenodo.4273220 [71].

**Ethics approval and consent to participate**
Not applicable

**Consent for publication**
Not applicable

**Competing interests**
The authors declare that they have no competing interests.

**Author details**
[1]Department of Computer Science, University of Illinois at Urbana-Champaign, Urbana, USA. [2]Department of Genetics, Stanford University, Stanford, USA. [3]Department of Molecular Pharmacology and Experimental Therapeutics, Mayo Clinic, Gonda 19-476, 200 First St SW, Rochester, MN 55905, USA. [4]Department of Computer Science, Carl R. Woese Institute of Genomic Biology, and Cancer Center of Illinois, University of Illinois at Urbana-Champaign, 2122, Siebel Center, 201 N. Goodwin Ave., Urbana, IL 61801, USA.

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

## 

