## [**Additional file 6.** Review History. · Genome Biology]

Review History

First round of review

Reviewer 1

Are you able to assess all statistics in the manuscript, including the appropriateness of statistical tests used? Yes, and I have assessed the statistics in my report.

Comments to author:

This is an interesting paper that applies a novel approach to understanding gene expression changes that are associated with, and may be causal for, cancer cells metastasizing. They use a colorectal cancer (CRC) cell line and select for increased invasiveness through a Matrigel matrix. They compare gene expression and several epigenetic marks between the initial cells, called M0, and those selected for 6 rounds, called M6 and noted extensive changes. The most interesting and innovative part of the paper is how they search for models for regulation of the metastasis process. They build probabilistic graphical models for the differentially expressed (DE) genes that consider occurrences of TF binding sites within various windows of the gene, and occurrences and changes in histone marks and DNA accessibility. Surprisingly, changes in DNA accessibility had almost no predictability (assessed by cross validation), whereas changes in histone marks provided the most information. They experimentally validated one of the predicted factors required for invasiveness, JunD. Overall I find the results to be interesting and the approach is demonstrated to provide useful information about the regulatory network involved in the process of metastasis. The method should be applicable in a wide range of studies.

I have a few suggested revisions that I think will help clarify certain issues.

1. More information about the choice of the 20 TFs used in the analysis. Were those the only ones with available ChIP-seq data, or were there other reasons for their choice?
2. I don't understand the reason for doing separate analyses for up and down regulation. Are the results better than if you combine them? It seems you should be able to generate an overall model that predicts both types of DE that would be more comprehensive.

Reviewer 2

Are you able to assess all statistics in the manuscript, including the appropriateness of statistical tests used? No, I do not feel adequately qualified to assess the statistics.

Comments to author:

With the rapid development of multiple omics sequencing approaches, there is a pressing need for algorithms which can integrate different omics information together. This study by Ghaffari et al, investigates the metastatic progress with multi-omics approach in CRC cells. They used an in vitro culture system to select cells with the ability to migrate and analyzed the changes in these cells at both transcriptomic level and epigenomic level. Combining their sequencing data with the publicly available ChIP-seq data of TFs, they found AP-1 complex is important for colon cancer cell migration and invasion.

In general, the concept of combing multi-omics information together to evaluate TF function is intriguing, while their algorithm which is largely built on the assumption that the process of functional TF regulating the expression of its targeting gene is always accompanied by the changes of multiple histone modifications is somehow problematic. The authors further used this assumption to pinpoint the regulations with high correlation with epigenetic changes, it will certainly filter out some other important regulation process without epigenetic changes. This modeling strategy is heavily relied on the published TF ChIP-seq data of the corresponding cell type (only 20 TFs were analyzed in this manuscript), which limits its generalizability. Also, the computational findings need to be better supported/strengthened with a greater set of well-crafted biological experiments.

Specific comments:

1. The authors focused on their modeling analyses with multi-omics sequencing data. However, it would be also necessary to represent how these histone markers and chromatin accessibility changed globally during this selection process. Is it showing a gradual change from M0-M2-M4 to M6 at epigenetic level? It is surprising to find ATAC-seq data is not helpful in this analysis, is it because the poor quality of ATAC-seq data or no big changes on chromatin accessibility during this process?
2. There is need to be very cautious when selecting a phenotype from a tumor cell line, since it shows strong genome instability and heterogeneous within the cell population. It would be necessary to also analyze the CNV and SNV information between early and late stage to see whether a specific genotype is enriched after the selection. Moreover, it would be also interesting to study that how does M6 gain higher invasiveness? Is it because the enrichment of a small cell population or some cells gain more invasiveness during selection? Is M6 a stable cell line or just a transit stage?
3. The authors need to clarify that JunD KD is only important for cell migration. It would be expected to see that cell proliferation is not affected in KD cells (eg: with a growth curve). It would be interesting to explore that how does JunD regulate cell migration. The authors should also prove that JunD enhances cell migration with overexpression assay.
4. How do the JunD binding sites change during the selection? How did the other components of AP-1 complex change after JunD KD?
5. In the PCA analyses of Fig S2, PC1 shows the major difference between early and late stage, but M4:2 is closely clustered with M6:2, and M4:1 is closely clustered with M6:1. Is it showing larger variations between biological replicates than different stages?
6. It would be helpful to mark some DEG genes in the volcano plot of Fig S3b, like CDH1, SMARCE1 targets and ESR1 targets.
7. The DEGs are mainly from the comparison between M0 and M6. It would be more interesting to analyze whether these DEGs are gradually changed during selection M0-M2-M4-M6 process. It would be informative to explore which genes are already changed as early as M2 stage (could be the driver), and which genes are only changed in late M6 stage.
8. The authors should clarify why is it more reliable for a H3K27ac change than a H3K4me1 change when overlapping with a TFBS, and how they determine the different evidentiary values for different types of dynamic histone markers.

9. The "KO" in Fig 5A should be "KD"?

10. Why does EZH2 show high weight in Fig 2D, even though it shows low score in Fig 2B and 2C?

11. Y axis labels should be added for Fig 3A, B, C, D, E, F, G.

Response to Reviewer 1

Comment: This is an interesting paper that applies a novel approach to understanding gene expression changes that are associated with, and may be causal for, cancer cells metastasizing. They use a colorectal cancer (CRC) cell line and select for increased invasiveness through a Matrigel matrix. They compare gene expression and several epigenetic marks between the initial cells, called M0, and those selected for 6 rounds, called M6 and noted extensive changes. The most interesting and innovative part of the paper is how they search for models for regulation of the metastasis process. They build probabilistic graphical models for the differentially expressed (DE) genes that consider occurrences of TF binding sites within various windows of the gene, and occurrences and changes in histone marks and DNA accessibility. Surprisingly, changes in DNA accessibility had almost no predictability (assessed by cross validation), whereas changes in histone marks provided the most information. They experimentally validated one of the predicted factors required for invasiveness, JunD. Overall, I find the results to be interesting and the approach is demonstrated to provide useful information about the regulatory network involved in the process of metastasis. The method should be applicable in a wide range of studies.

Response: We thank the reviewer for their favorable comments. As noted by the reviewer, our method can indeed be used in a variety of studies and we appreciate the reviewer's recognition of this aspect.

Comment: 1. More information about the choice of the 20 TFs used in the analysis. Were those the only ones with available ChIP-seq data, or were there other reasons for their choice?

Response: The ENCODE database includes ChIP-seq binding data for 24 TFs in the HCT116 cell line (chosen because it is also a colorectal cancer cell line); four of these 24 ChIP profiles had low read depth. Therefore, we included all remaining 20 TFs in our analyses. We have now added the following clarification to the Methods section:

“(These included all 20 TFs for which the available ChIP-seq profiles had high read depth.)”

Comment: 2. I don't understand the reason for doing separate analyses for up and down regulation. Are the results better than if you combine them? It seems you should be able to generate an overall model that predicts both types of DE that would be more comprehensive.

Response: The reviewer raises an interesting question. Our reasons for separating the analysis of up- and down-regulated genes were related to an aspect of the pGENMi model, which we explain below in detail beyond that permitted in the manuscript; this in turn implied that the “combined” modeling was expected to be less useful/powerful than the separate analyses, which we have since demonstrated empirically. A summary of this rationale has also been added to the manuscript.

The reason for doing separate analyses for up and down regulation is the following. The pGENMi model attempts to explain p-values of differential expression in terms of TF-related regulatory evidences. These p-values do not contain information on the directionality of the expression changes. As such, in a “combined” analysis, the model will attempt to use the same cis-regulatory evidence, with the same weight, to explain both up- and down-regulation simultaneously; the resulting weights will be less reliable. If the original model were written so as to explain, say, the

z-score of a gene's expression (which is positive or negative depending on whether the gene is up- or down-regulated), this would not have been a problem. However, such a formulation of the model would have required substantial changes from pGENMi, and a number of additional tests, making it outside the scope of this work.

Thus, our decision to separate the up- and down-regulation analysis was based on the above theoretical reasoning. That being said, to demonstrate the undesirable effects of “combining” the two analyses, we carried out and now report an additional analysis. Recall that in our original analysis, each analysis (whether for up- or down-regulation) examined all genes' p-values (of differential expression); however, when analyzing, say up-regulation, we converted the p-values of differential expression into p-values of up-regulation. (Thus, a p-value close to 0 for a down-regulated gene would be assigned a p-value close to 1 in the up-analysis, etc.) Now, we performed “fw-pGENMi” model training with p-values that correspond to either direction of dysregulation. We compared the learned weights of TFs and dynamic histone marks with corresponding weights trained with up-regulation p-values only (known as up-analysis, already provided in the original manuscript **Figures 2 d,e**). As shown in **Figure R1b** below, all the histone mark weights in the new combined analysis are negative, whereas in the separated analysis (in the original report) the weights associated with a histone mark going up or down had opposite signs. For example, up-analysis learns a positive weight for K27ac_up (i.e., an increase in this mark at an activator site is evidence of up-regulation) and a negative weight for K27ac_down (i.e., a decrease in this mark at an activator site is counter-evidence of up-regulation). While these learnt weights from the separate analyses agree with our prior beliefs, the combined analysis learns negative weights for both direction of changes in K27ac. (This suggested to us that the down-regulated genes had a more dominant effect in training the weights.) Similar observations can be made when comparing weights learned for TFs (**Figure R1a** below). For example the TF ZFX was found in our original analyses to have a strong contribution to the models for up- as well as down-regulation (**Figures 2c,b** in the original submission) and was assigned a strong positive weight in both the down- (highest among all 20 TFs) and up- (5th highest) analysis (**Figure 2d**). However, the combined analysis learned a weak positive weight for this TF.

We have added the theoretical motivation for doing the analyses separately to the revised manuscript, in Discussion, page 21. The added text is reproduced below as well.

“Another technical point worth noting is that our modeling was performed separately for up-regulation and down-regulation between stages. The pGENMi model attempts to explain p-values of differential expression, but these p-values do not contain information on the directionality of the expression changes. As such, in a “combined” analysis, where both types of differential expression p-values are present, the model attempts to use the same cis-regulatory evidence, with the same weight, to explain both up- and down-regulation and will be confounded; the resulting weights will be less reliable. This is the primary reason why we separated the up- and down-analyses.”

Figure R1. Comparison of “combined” analysis weights with up-analysis weights. Weights learned by fw-pGENMi for TFs (a) and dynamic histone marks (b) using differential expression p-values that represent up- as well as down-regulation (“combined analysis”) versus corresponding weights learned in up-analysis. For the combined analysis weights to be comparable with up-analysis weights, we trained fw-pGENMi using the same distance threshold as that used for up-analysis, i.e. 50Kb, and only tuned the regularization coefficient using cross-validation.

Response to Reviewer 2

Comment: With the rapid development of multiple omics sequencing approaches, there is a pressing need for algorithms which can integrate different omics information together. This study by Ghaffari et al, investigates the metastatic progress with multi-omics approach in CRC cells. They used an in vitro culture system to select cells with the ability to migrate and analyzed the changes in these cells at both transcriptomic level and epigenomic level. Combining their sequencing data with the publicly available ChIP-seq data of TFs, they found AP-1 complex is important for colon cancer cell migration and invasion.

In general, the concept of combing multi-omics information together to evaluate TF function is intriguing, while their algorithm which is largely built on the assumption that the process of functional TF regulating the expression of its targeting gene is always accompanied by the changes of multiple histone modifications is somehow problematic.

Response: We thank the reviewer for their thoughtful and constructive comments. With regard to the relationships between TF-regulated expression and histone state, we believe that such changes within/at TF binding sites, when present, bolster the evidence in support of the binding site being functional. Intuitively speaking, this is a strategy that is expected to increase precision in identification of functional sites, potentially at the expense of sensitivity; but since the model's primary goal is to identify major TFs, which it does by aggregating evidence over many binding sites of each TF, the above precision-sensitivity tradeoff works to improve the overall accuracy of the model. We demonstrated this by objective comparisons to a version of the approach where TF binding sites are used as evidence of regulation without regard to epigenomic information (**Figures 3b-e**).

We also wish to clarify that the model is not reliant on *multiple* changes to histone modifications occurring at a given TF binding site. The evidence of functionality is allowed to be borne out by *one or more* modifications, which the model does not presume to know in advance. A particular binding site of a TF may show only one epigenomic change, while another may show a different epigenomic change, and the model learns to account for these different types of epigenomic evidences without requiring both or all of them to be present at any given binding site. The model also allows supporting evidence to come from say one or two histone modifications for some TF(s) and from several different histone modifications for other TF(s).

To illustrate the above point, we examined the weights learned by the pGENMi model (in its default setting). **Figure R2** below shows a heatmap of the weights learned in the analysis of up-regulation, where each row contains the weights associated with a TF in combination with each dynamic epigenomic mark. Darker and lighter colors represent more and less significant weights respectively. Roughly speaking, a strong weight (say > 0.2 in absolute value) means that a relatively large subset of DE genes is enriched in the binding site of the TF overlapping with a mark changing in a specific direction whereas such evidence is found less abundantly near non-DE genes. In other words, an epigenomic change with a strong weight in the row for a TF counts as reliable evidence of functionality of the TF's binding site. For a TF to be predicted as important in progression the presence of at least one strong weight is necessary, again roughly speaking; the presence of multiple strong weights is not required, though it is expected that TFs with multiple strong weights gain higher ranks in the final ranking provided in **Figure 2b,c**. For example, POLR2A and FOSL1 are ranked 6 and 7 respectively in **Figure 2c** and have one strong weight whereas CTCF, JUND, ZFX, and RAD21 have multiple significant weights.

Moreover, we had reported on a variant strategy (“DiffMarkAggr”) where the presence of *any* mark changing within a TF binding site is treated as evidence of that binding site functionally regulating the gene. This strategy does not require changes to multiple histone modifications at a binding site and has the second-best and third-best performances among the evaluated strategies in down and up-analysis, respectively, as plotted in **Figure 3b,c**. This strategy therefore has merits to it, but the same figure also showed that the default strategy of keeping track of specific histone modifications is better at explaining and predicting differential expression.

Figure R2: Heatmap of weights associated with each combination of TF and dynamic epigenomic mark. Weights were learned by pGENMi analysis of up-regulation p-values. Strong weights (e.g., greater than 0.2 in absolute value) indicate epigenomic changes that the model considers as evidence of functionality of a specific TF’s binding sites.

Comment: The authors further used this assumption to pinpoint the regulations with high correlation with epigenetic changes, it will certainly filter out some other important regulation process without epigenetic changes.

Response: The reviewer makes a valid point, and we agree. There is a tradeoff between sensitivity and precision at the level of detecting functional binding sites, and the model favors precision. However, as we noted above, predicting TF binding sites is not the primary goal but only the means to identify significant TFs underlying the biological process. We demonstrated systematically that this strategy improves the model's ability to meet this goal. In particular, we evaluated a baseline strategy ("TFBS-only") where the presence of a TFBS near the gene is considered as the TF's regulatory evidence, without regard to epigenomic evidence. The fundamental issue with TFBS-only analysis is that it is confounded by a high-rate of false-positive sites (in prediction of binding sites), a problem that is exacerbated when searching over longer intergenic regions. As a result, in our objective evaluations based on cross validation (**Figure 3b, c**), the TFBS-only strategy had substantially worse accuracy in terms of explaining the data compared to the default strategy (DiffMark).

Comment: This modeling strategy is heavily relied on the published TF ChIP-seq data of the corresponding cell type (only 20 TFs were analyzed in this manuscript), which limits its generalizability.

Response: We do not see our use of published ChIP-seq data of the appropriate cell type as a limitation but a strength. **Firstly**, only 20 TFs (ChIP-seq profiles) were analyzed because these were all the ChIP profiles (of high read depth) that were available from ENCODE for the colorectal cancer cell line HCT116. (Four TF profiles were rejected by us due to low depth.) **Secondly**, the main goal of the work was to demonstrate how ChIP-seq profiles can be combined with dynamic epigenomic profiles to learn important regulators and regulatory interactions in a particular biological process; we do not see how we could have avoided relying on published TF ChIP-seq data. **Thirdly**, the model can in fact be provided with data on any number of TFs. To showcase such capability, we repeated DiffMark analysis using 216 TF binding data from K562 cell line. The rankings of top 20 K562 TFs for down- and up-analyses are shown in **Figure R3a,b**. We noted some of these top TFs to be common to the top TFs from the original analysis performed using HCT116 ChIP-seq data. For example, FOSL1 and JUND (members of the AP1 complex) appear at ranks 1 and 14 in the down-analysis (**Figure R3a**); both were among the top five TFs in the corresponding analysis with HCT116 data. **Fourthly**, in case experimental binding data for the same cell line are not available, our work suggests that the use of TF ChIP profiles from a different cell type can be substituted, but with an accompanying loss of predictive capacity. Our evaluations (**Figure 3f,g**, reproduced below as **Figure R4** for the reviewer's convenience) show that such a scheme also has strong predictive power (red distributions in figure; ChIP data from K562 cell line) though not as strong as when using ChIP data from the appropriate cell type (black, dashed vertical lines; ChIP data from HCT116). This finding is reasonable, since TF ChIP profiles often have significant levels of similarity between cell types, reflecting the sequence-dependent aspect of binding which remains unchanged. **Finally**, we note that if no ChIP data are available, sequence-based approaches to predict binding using TF motifs and epigenomic data (e.g., CENTIPEDE, PMID: 21106904) can also be used. Thus, in light of the above arguments, we do not believe the modeling strategy is limited in generalizability.

Figure R3. Training pGENMi with 216 TF ChIP data from K562 cell line. pGENMi was trained using 216 TFs from K562 cell line in DiffMark strategy. The top 20 TFs in the ranking of 216 TFs are shown in (a) and (b) for down- and up-analysis respectively.

Figure R4. Figure 3.f, g from the original manuscript, reproduced here for convenience of the reviewer.

Comment: Also, the computational findings need to be better supported/strengthened with a greater set of well-crafted biological experiments.

Response: We have provided our responses to this general comment in the context of specific comments noted below.

Comment: 1. The authors focused on their modeling analyses with multi-omics sequencing data. However, it would be also necessary to represent how these histone markers and chromatin accessibility changed globally during this selection process. Is it showing a gradual change from M0-M2-M4 to M6 at epigenetic level?

Response: We agree with the reviewer that exploratory analysis for histone mark and chromatin accessibility changes can provide a high-level description of the data and biological process. We have now added figures to the manuscript that address this aspect.

First, we examined how counts of histone mark ChIP peaks and DNA accessibility peaks change across stages. To make the examination more informative for our purposes, we considered peaks within 10Kb upstream of genes for each stage of progression, limiting ourselves to genes that are differentially expressed (p-value < 0.05) between early (M0) and late (M6) stages. (Up- and down-regulated genes were examined separately, for consistency with other analyses in the manuscript.) The results are shown in **Figure R5** below which has also been added to the manuscript as **Figure S5**. For instance, we see that H3K27ac peaks near down-regulated genes are fewer in later stages and those near up-regulated genes are more numerous in later stages, with the count increasing gradually through the stages. (The trends are in line with expectation from this activating histone mark.) The reverse pattern exists for H3K27me3 peaks, consistent with a repressive role of this mark. Accessibility peaks near up-regulated genes are slightly more numerous in the later stage and those near down-regulated genes are slightly less numerous in the later stage, consistent with expectation.

Next, we explored signal strengths between M0 and M6, for DE genes. **Figure R6** below shows the maximum height of peaks located within 10 Kbp upstream of each DE gene, for M6 versus M0. The plots were generated for up and down-regulated genes separately, to be consistent with the analyses in the manuscript. For the activating mark H3K27ac, there is an increase in peak height in the later stage, for a majority of up-regulated genes and a decrease in peak height for the majority of down-regulated genes. The same pattern exists for H3K4me1 and H3K4me3 as they are activating marks, and the opposite pattern exists for K27me3 as it is a repressive mark. On the other hand, accessibility peaks do not show a clear trend of greater or lesser strength in one stage versus the other; there are similar numbers of peaks whose strength increases or decreases in the later stage.

In short, the epigenomic changes through the stages tend to be gradual, though there are also examples where a sharp change in peak count or a frequent change in peak height is observed. We have now added the **Figures R5** and **R6** as supplementary **Figures S5** and **S6**, and the following description of the explanatory analysis has been added to the manuscript in Results, page 8.

“We first examined global changes in histone marks and chromatin accessibility by summarizing how counts of histone mark ChIP peaks and DNA accessibility peaks change across stages. Specifically, we counted peaks within 10 Kbp upstream of genes for each stage of progression, limiting ourselves to genes that are differentially expressed (p-value < 0.05) between early (M0) and late (M6) stages. This was done for up- and down-regulated genes separately. The results (**Additional file 1: Figure S5**) show clear trends of genome-wide epigenomic changes. For instance, H3K27ac peaks near down-regulated genes are fewer in later stages and those near up-regulated genes are more numerous in later stages, as might be expected of an activating histone mark. The reverse pattern exists for H3K27me3 peaks, consistent with a repressive role for this mark. Similar trends were observed in changes of signal strength between stages (**Additional file 1: Figure S6**).”

Figure R5: Histone mark and chromatin accessibility changes during progression. For each mark and chromatin accessibility, the count of peaks within 10Kb upstream of the DE genes (DE p-value < 0.05) is shown for each stage, separately for down-regulated (left) and up-regulated (right) genes. For H3K27ac there is a sharp decrease in the peak count from M0, M2 to M4, M6 and for H3K27me3 there is a sharp increase in the peak count from M0, M2 to M4, M6 near down-regulated genes. There is a gradual increase and a gradual decrease in the peak counts near up-regulated genes for H3K27ac and H3K27me3 respectively. For H3K4me3, there is a monotonic

decrease from M2 to M6 and a monotonic increase from M0 to M4 in the peak counts near down and up-regulated genes respectively. Also, comparison of M0 with M6 shows that the peak counts for H3K4me3 increase slightly for up-regulated genes and decrease for down-regulated genes. For H3K4me1 the peak counts monotonically increase near up-regulated genes, whereas near down-regulated genes, the number of peaks goes up for M2 then goes down and there is an increase in the peak count for M6 vs M0. The number of accessible regions does not change substantially between M0 and M6, for either set of DE genes. There is a slight increase and decrease in the number of accessible sites for up- and down-regulated genes respectively. For easier visualization of these minor changes only the count of peaks with height of at least 500 are represented in the plot.

Figure R6. Maximum height of peaks within 10Kb upstream of each DE gene for M6 versus M0. Each plot represents how the maximum height of a histone mark ChIP peak or accessibility (ATAC-seq) peak within 10Kb upstream of each DE gene (“DEG”, DE p-value < 0.05) changes in M6 vs M0. The left and right plots show this information for down- and up-regulated genes respectively and the insets show the count of the genes above or below the identity line. For all

activating histone marks there is an increase in peak height near the majority of up-regulated genes whereas there is a decrease in the peak height near the majority of down-regulated genes. The opposite pattern exists for H3K27me3, which is believed to be a repressive mark. For accessibility peaks, there is no significant difference between the number of DE genes on either side of the identity line, suggesting that accessibility strength does not change predominantly in one direction over the other.

Comment: It is surprising to find ATAC-seq data is not helpful in this analysis, is it because the poor quality of ATAC-seq data or no big changes on chromatin accessibility during this process?

Response: **Figure R5** above (bottom panel) suggests that the numbers of chromatin accessibility peaks do not change significantly during the process. (For a more focused examination of this point, we counted peaks near up- and down-regulated genes.) **Figure R6** above (bottom panel) suggests that the peak strength changes in either direction (increase or decrease) with similar frequency, even when looking at differential expression in one direction (e.g., genes up-regulated in later stage). These simple visualizations offer clues about why change of accessibility may not be a strong marker of up-regulation or down-regulation in this context, and thus offers an explanation of our finding that ATAC-seq data are not as useful as histone modifications for the model.

We agree with the reviewer's sentiments that qualities associated with ATAC sequencing may have also contributed to the finding that ATAC-seq data was not as informative as specific ChIP-seq data in these analyses. For example, ATAC-seq "peaks" tend to be broader and less well-defined than those for most ChIP-seq data, e.g., H3K27ac, because of the transposon-based nature of ATAC. Therefore, analysis tools appropriate for these data structures were used in our analyses (see methods). The wider peaks (hills/islands) could also lessen specificity for a given TF within an analysis window. We would also point out that our data do not indicate that ATAC-seq data are not informative in our model, only that certain histone marks, assessed by ChIP-seq, appear to be more informative. HTS read quality itself does not appear to have been an issue with ATAC data. Following adapter trimming, sequence quality pruning, mapping and deduplication, ATAC libraries consisted of 27–42 million mapped reads each.

Comment: There is need to be very cautious when selecting a phenotype from a tumor cell line, since it shows strong genome instability and heterogeneous within the cell population. It would be necessary to also analyze the CNV and SNV information between early and late stage to see whether a specific genotype is enriched after the selection.

Response: The reviewer is correct that culture heterogeneity and genome instability likely both contribute to the predominant genomes in a culture derived via selection, similar to how these factors determine the outcome of selective pressures in various tumor processes. To directly address this point, variant calling was performed using existing RNAseq data for M0 and M6 cultures. These data have been added to manuscript as **Figure S4** and are included below as **Figure R7**. Descriptive text related to the new figure has been added to page 7. A subset of variant loci (single nucleotide variants with high depth across M0 and M6 lines) was selected to assess shifts in population allele frequencies between M0 and M6 as a measure of enrichment. The clear trend in these plots is that allele frequencies are by and large similar between the stages, with a relatively small number of exceptions (points along the axes in **Figure R7**).

Full characterization of all genotype changes that accumulated between M0 and M6 is beyond the scope of this manuscript, which is focused on epigenomic changes at TF binding sites; however, the additional analyses confirm that the genetic cell identity of the M6 population is similar to that of the M0 (SW480) population. Among the 205 protein-coding genes harboring the SNVs with significant differences in alternative allele frequency (right panels in **Figure R7**), no obvious driver mutations associated with CRC progression were noted (**Additional file 3**).

Figure R7. Changes in alternative allele frequencies between early and late stages. Average alternative allele frequency for M6 vs M0 for all the SNVs ($n = 1516$) having sufficient read depth (average read depth for all SNVs was used as threshold) (a), and the subset of SNVs ($n = 674$) that exhibit significant change (aggregated binomial test p -value < 0.05) in alternative allele

frequency from M0 to M6 **(b)**. The significance of change was determined by testing the deviation of alternative allele frequency for each replicate of M6 stage from the average M0 alternative allele frequency using binomial test and aggregating p-values using Fisher's method. **(c)** and **(d)** are 2D histograms associated with **(a)** and **(b)** respectively.

Comment: Moreover, it would be also interesting to study that how does M6 gain higher invasiveness? Is it because the enrichment of a small cell population or some cells gain more invasiveness during selection? Is M6 a stable cell line or just a transit stage?

Response: We agree with the reviewer, and we are actively investigating the specific mechanisms through which invasiveness is being driven in M6 cells. While outside of the scope of this manuscript, we hypothesize that, similar to what is observed in highly heterogeneous tumors, multiple factors ultimately contribute to the invasive phenotype. These include genetic heterogeneity within the cell population itself (e.g., see previous comment and response), as well as heterogeneity within the epigenomes and transcriptomes. Furthermore, dynamic interplay between these mechanisms (e.g., epigenetic changes driving increased mutation rates and/or the inverse, with mutations driving epigenetic changes) possibly contributes. While outside of the scope of the manuscript, to directly address the reviewer's questions regarding the stability of the M6 invasive phenotype, we have observed consistently high invasiveness across prolonged culture (in the absence of selective pressure). Future studies will be directed at determining if the gene expression and epigenetic programs that support this phenotype are stable at baseline (i.e. while not actively invading) or if the programs are activated during the invasion process (i.e. plasticity is the predominate mechanism). New models and analytical tools outside of the scope of the present study are being developed to aid in these advanced analyses. It is also noted that we did not see new mutations in any clear metastatic driver genes (discussed in previous responses) and our data indicate that large-scale changes are unlikely as they would have manifested as an accumulation of data near the coordinates $x=1,y=0$ and $x=0,y=1$ in the **Figure R7** included with the response to the previous question. The analytical frameworks we detail in this manuscript will be instrumental in deciphering these vexing questions as we continue to investigate these phenomena in cell and patient derived models of cancer progression.

Comment: The authors need to clarify that JunD KD is only important for cell migration. It would be expected to see that cell proliferation is not affected in KD cells (eg: with a growth curve). It would be interesting to explore that how does JunD regulate cell migration. The authors should also prove that JunD enhances cell migration with overexpression assay.

Response: We have added new data to the revised manuscript showing that proliferation is unaffected by JunD knockdown, as requested (see **Figure S15** included below as **Figure R8**). The data are also described on page 17 of the revised manuscript. We agree that additional investigation into JunD-mediated regulation of cell migration will be interesting for future studies; however, such in-depth gene-focused studies are outside of the scope of the present manuscript. It is noted that our data supported JunD enhances invasiveness through epigenomic activation of specific JunD binding sites (e.g., please see **Figures 2b-2d**). Given this, it would be expected that knockdown of JunD would elicit an effect on those sites (e.g., removing JunD's ability to bind the site by removing JunD itself). On the other hand, overexpression of JunD in this situation would not necessarily mean that JunD was capable of binding to epigenetically repressed TF binding sites to enhance migration. If overexpressed JunD activated these repressed sites, it

would be a separate mechanism that would be similar to pioneer transcription factor-like processes and would break epigenetic control mechanisms (i.e. it would be effectively “overloading the system” instead of addressing the question of TF accessibility). We felt that the overexpression approach would not be specific to the differentially accessible sites identified in our study and therefore would not appropriately address our hypothesis.

Figure R8. Cell proliferation is unaffected by JunD knockdown. Proliferation was measured for SW480 M0 (a) and M6 (b) cell lines following shRNA-mediated knockdown of JunD (JunD) or using a non-targeting control (Scr). Solid lines represent the mean of 3 independent wells assayed in parallel. Dashed lines represent the standard deviation.

Comment: How do the JunD binding sites change during the selection?

Response: To address the reviewer’s question, we asked how the number of peaks for each histone mark and chromatin accessibility change within JunD binding sites located within 50Kb upstream or downstream of protein coding genes (Figure R9). We examined sites within 50 Kbp for consistency with the main analysis of the paper, where this distance threshold was determined to be optimal. We examined JunD sites vis-à-vis their overlap with epigenomic information since the binding data (ChIP-seq) on JunD are from the ENCODE project (related cell line HCT116) and do not provide information on change between stages.

In Figure R9, the red and green bars indicate the number of JunD sites overlapping epigenomic peaks exclusive to M6 and M0 respectively, and the yellow bar shows the number of JunD sites overlapping epigenomic peaks common to both stages. We noted that JunD sites show increase as well as decrease of H3K27ac modifications in roughly equal proportions; the same trend is seen for chromatin accessibility. On the other hand, JunD sites harbor disproportionate instances of H3K4me1 modifications exclusive to the later stage. This is consistent with the model’s finding that an increase (resp. decrease) of H3K4me1 modification at JUND sites is highly informative of

up- (resp. down-) regulation of genes. For chromatin accessibility, H3K4me3 and H3K27ac, the most frequent observation at JunD sites is that of no change, i.e., the epigenomic modification at these sites is found in both stages. These observations give us a high-level perspective on the kinds of changes seen at JunD sites between the early and late stages. At the same time, they motivate a more nuanced examination of the changes in light of gene expression changes, which is precisely what our model and analysis attempts to do.

We have now added **Figure R9** as Supplementary **Figure S8**, and the following text has been added to the manuscript in Results, page 10.

“A direct look at the binding sites of one of these TFs, viz., JunD (**Additional file 1: Figure S8**), shows substantial epigenomic changes in both directions. However, it does not immediately offer a mechanistic explanation of such changes or a quantitative assessment of their impact on gene expression and illustrates the need for a more nuanced analysis cognizant of expression changes, as is provided by our model.”

Figure R9. Changes in the number of histone mark and accessibility peaks within JunD binding sites. For each histone mark and for accessibility, the count of the peaks within JunD binding sites, located within 50Kb upstream or downstream of the protein coding genes, is partitioned into three categories. The red bars show the number of peaks exclusive to M6, the green bars show the number of peaks exclusive to M0, and the yellow bars indicate the number of peaks shared between the two stages. The number of peaks exclusive to M0 is almost equal to the number of exclusive peaks to M6 for H3K27ac and accessibility (ACC). The number of peaks exclusive to M6 is much larger than the number of peaks exclusive to M0 for H3K4me1 and the opposite pattern exists for H3K4me3. For H3K27me3 the total number of peaks is small, and the total number of exclusive peaks is larger than the common peaks. Accessibility and H3K4me1 have the largest and smallest ratio of the number of common sites to the total number of exclusive sites.

Comment: How did the other components of AP-1 complex change after JunD KD?

Response: We examined expression changes in the other components of the AP1 complex upon JunD knock-down (**Table R1** below). Using a q-value cutoff of 0.1, only FOSL1 and JunD are significantly down-regulated and most of the components do not change expression significantly. Stronger knockdown at the protein level was observed than at the mRNA level (e.g., see **Figure 4a**), indicating that translation was inhibited by the shRNA, but the transcript was not necessarily cleaved. We have now added this table to the Supplement as **Table S3**. Descriptive text related to the new table has been added to page 17.

Gene	FC	p-value	q-value
ATF6B	1.003	0.5	1.0
FOSL1	0.64	0.00004	0.002
ATF7	1.6	0.1	0.5
FOS	1.05	0.3	0.8
BATF	0.84	0.07	0.3
JUN	0.95	0.2	0.6
MAFG	0.99	0.4	0.9
JUNB	0.98	0.3	0.8
JUND	0.78	0.02	0.1
ATF5	1.2	0.07	0.3

Table R1. Expression changes in AP1-complex components upon JunD Knockdown.

Comment: 5. In the PCA analyses of Fig S2, PC1 shows the major difference between early and late stage, but M4:2 is closely clustered with M6:2, and M4:1 is closely clustered with M6:1. Is it showing larger variations between biological replicates than different stages?

Response: Yes, the reviewer is correct in their interpretation. As explained in the “Results” section of the manuscript, PCA analysis indicates that the expression profiles for M4 and M6 are distinct from those for M0 and M2 (**Additional file 1: Figure S2**), but profiles for M4 and M6 are more similar within replicates than by stage. Thus, M0 and M6 were chosen for further analyses because of the clear expression and phenotypic separation between them.

Comment: 6. It would be helpful to mark some DEG genes in the volcano plot of Fig S3b, like CDH1, SMARCE1 targets and ESR1 targets.

Response: Figure S3 has been adjusted in the revised manuscript to mark DEG genes in the pathways indicated in the results section, which includes CDH1, SMRCE1, and ESR1 targets. We reproduce it below for the reviewer's convenience (**Figure R10**).

Figure R10. Figure S3 from the original manuscript, reproduced here for convenience of the reviewer.

Comment: 7. The DEGs are mainly from the comparison between M0 and M6. It would be more interesting to analyze whether these DEGs are gradually changed during selection M0-M2-M4-M6 process. It would be informative to explore which genes are already changed as early as M2 stage (could be the driver), and which genes are only changed in late M6 stage.

Response: As noted above, we did not judge the expression data to reliably distinguish between M0 and M2. Examination of transcriptional changes at a higher resolution is an excellent topic for future work.

Comment: 8. The authors should clarify why is it more reliable for a H3K27ac change than a H3K4me1 change when overlapping with a TFBS, and how they determine the different evidentiary values for different types of dynamic histone markers.

Response: Our model was specifically designed to automatically learn a weight for each dynamic histone mark that reflects its relative value as an evidence for TFBS function. Explanation of the model was provided in Results (pages 9, and 10) and Methods (page 27). As an example, consider all the TFBS of a particular TF, say JunD, in the regulatory region of genes. Divide these into those that overlap with a H3K27ac mark that increases in the later stages, and those that do not have such overlap. Determine the extent to which these two subsets separate DE genes from non-DE genes non-randomly. This is an approximate measure of how reliable H3K27ac_up is as evidence of function when overlapping a binding site. Then repeat this exercise with H3K4me1 to assess its reliability. By comparing these, one will usually find one mark to be more reliable than another. This is similar to how the model operates, except that it does the assessment of reliability in a more integrated manner (considers all TFs' sites and all histone marks in a "multiple regression" sort of way) and without imposing any arbitrary thresholds on DE p-values.

Comment: 9. The "KO" in Fig 5A should be "KD"?

Response: We thank the reviewer for spotting the error. This is now fixed.

Comment: 10. Why does EZH2 show high weight in Fig 2D, even though it shows low score in Fig 2B and 2C?

Response: There are two main differences between what is shown in **Figure 2d** and in **Figures 2b,c**. Firstly, the TF rankings in **Figure 2b,c** are based on each TF's contribution in modeling the data, as measured by change in overall goodness-of-fit (log likelihood) of the entire data set when that TF is removed from the model. In other words, these rankings are based on the $\Delta(\text{LLR})$ score of a TF. On the other hand, the TF scores shown in **Figure 2d** are the weights associated with those TFs in the model, reflecting the strength (how reliable) and direction (activating/repressive) of evidence presented by a binding site of that TF. The two scores are statistically related to each other, e.g., if the weight of a TF is learnt to be close to zero, its contribution to the model ($\Delta(\text{LLR})$) will generally be small. However, these two scores are also distinct, e.g., a TF's binding site may be a reliable predictor of up-regulation when it is seen near a gene, making its weight high, but it may be seen near relatively few genes and thus may not make a major contribution to the model's ability to explain the entire data set. Thus, we were not surprised that there are cases such as EZH2 where the weight is learnt to be strong (**Figure 2d**) but the $\Delta(\text{LLR})$ is modest (low rank in **Figures 2b,c**).

A second difference between those two scores is that they were obtained from different modeling exercises – **Figure 2b,c** are from the pGENMi model and **Figure 2d** is from the fw-pGENMi model. (As explained in Results, pages 11 and 12, the fw-pGENMi model is a simpler model that ensures consistency of evidentiary values of histone marks and TFs.) To respond to the reviewer’s question, we now repeated the TF ranking exercise (**Figure 2b,c**) with the fw-pGENMi model; these are shown below in **Figures R11a,b**. EZH2 is now seen to have a middle rank (9) in down-analysis and high rank (5) in up-analysis.

Figure R11. Ranking of TFs by their contribution to model ($\Delta(\text{LLR})$) in down- and up-analysis respectively using fw-pGENMi.

Comment: 11. Y axis labels should be added for Fig 3A, B, C, D, E, F, G.

Response: Y-axis labels have been added for Figures 3a, b, c, d, e, f, and g.

Second round

of review

Reviewer 1

I have no further questions or revisions to suggest.

Reviewer 2

The authors have well addressed most of my questions and comments.